# Sub-sampling for Efficient Non-Parametric Bandit Exploration

**Dorian Baudry**
dorian.baudry@inria.fr

**Emilie Kaufmann**
emilie.kaufmann@univ-lille.fr

**Odalric-Ambrym Maillard**
odalric.maillard@inria.fr

Univ. Lille, CNRS, Inria, Centrale Lille, UMR 9198-CRIStAL, F-59000 Lille, France

## Abstract

In this paper we propose the first multi-armed bandit algorithm based on *re-sampling* that achieves asymptotically optimal regret simultaneously for different families of arms (namely Bernoulli, Gaussian and Poisson distributions). Unlike Thompson Sampling which requires to specify a different prior to be optimal in each case, our proposal RB-SDA does not need any distribution-dependent tuning. RB-SDA belongs to the family of Sub-sampling Duelling Algorithms (SDA) which combines the *sub-sampling* idea first used by the BESA [1] and SSMC [2] algorithms with different sub-sampling schemes. In particular, RB-SDA uses *Random Block* sampling. We perform an experimental study assessing the flexibility and robustness of this promising novel approach for exploration in bandit models.

## 1 Introduction

A $K$-armed bandit problem is a sequential decision-making problem in which a learner sequentially samples from $K$ unknown distributions called arms. In each round the learner chooses an arm $A_t \in \{1, \dots, K\}$ and obtains a random reward $X_t$ drawn from the distribution of the chosen arm, that has mean $\mu_{A_t}$. The learner should adjust her sequential sampling strategy $\mathcal{A} = (A_t)_{t \in \mathbb{N}}$ (or bandit algorithm) in order to maximize the expected sum of rewards obtained after $T$ selections. This is equivalent to minimizing the *regret*, defined as the difference between the expected total reward of an oracle strategy always selecting the arm with largest mean $\mu_\star$ and that of the algorithm:

$$\mathcal{R}_T(\mathcal{A}) = \mu_\star T - \mathbb{E}\left[\sum_{t=1}^{T} X_t\right] = \mathbb{E}\left[\sum_{t=1}^{T} (\mu_\star - \mu_{A_t})\right].$$

An algorithm with small regret needs to balance exploration (gain information about arms that have not been sampled a lot) and exploitation (select arms that look promising based on the available information). Many approaches have been proposed to solve this exploration-exploitation dilemma (see [3] for a survey), the most popular being Upper Confidence Bounds (UCB) algorithms [4, 5, 6] and Thompson Sampling (TS) [7, 8]. TS is a randomized Bayesian algorithm that selects arms according to their posterior probability of being optimal. These algorithms enjoy logarithmic regret under some assumptions on the arms, and some of them are even *asymptotically optimal* in that they attain the smallest possible asymptotic regret given by the lower bound of Lai & Robbins [9], for some parametric families of distributions. For distributions that are continuously parameterized by their means, this lower bound states that under any uniformly efficient algorithm,

$$\liminf_{T \to \infty} \frac{\mathcal{R}_T(\mathcal{A})}{\log(T)} \geq \sum_{k:\mu_k < \mu_\star} \frac{(\mu_\star - \mu_k)}{\mathrm{kl}(\mu_k, \mu_\star)}., \tag{1}$$

where $\mathrm{kl}(\mu, \mu')$ is the Kullback-Leibler divergence between the distribution of mean $\mu$ and that of mean $\mu'$ in the considered family of distributions. For arms that belong to a one-parameter exponential

family (e.g. Bernoulli, Gaussian, Poisson arms) kl-UCB using an *appropriate divergence function* [6] and Thompson Sampling using an *appropriate prior distribution* [10, 11, 12] are both *asymptotically optimal* in the sense that their regret matches that prescribed by the lower bound (1), for large values of $T$. Yet, a major drawback of theses algorithms is that their optimal tuning requires the knowledge of the families of distributions they operate on. In this paper, we overcome this issue and propose an algorithm that is simultaneously asymptotically optimal for several families of distributions.

In the past years, there has been a surge of interest in the design of non-parametric algorithms that directly use the empirical distribution of the data instead of trying to fit it in an already defined model, and are therefore good candidates to meet our goal. In [13], the authors propose the General Randomized Exploration (GRE) framework in which each arm $k$ is assigned an index $\hat{\mu}_{k,t}$ sampled from a distribution $p(\mathcal{H}_{k,t})$ that depends on the history of past observed rewards for this arm $\mathcal{H}_{k,t}$, and the arm with largest index is selected. GRE includes Thompson Sampling (for which $p(\mathcal{H}_{k,t})$ is the posterior distribution given a specified prior) but also allows for more general non-parametric re-sampling schemes. However, the authors of [13, 14] show that setting $p(\mathcal{H}_{k,t})$ to be the non-parametric Bootstrap [15] leads to linear regret. They propose variants called GIRO and PHE which perturb the history by augmenting it with fake samples. History perturbation was already suggested by [16] and is also used by Reboot [17], with a slightly more complicated bootstrapping scheme. Finally, the recently proposed Non Parametric TS [18] does not use history perturbation but instead sets $\hat{\mu}_{k,t}$ as a weighted combination of all observations in $\mathcal{H}_{k,t}$ and the upper bound of the support, where the weights are chosen uniformly at random in the simplex of dimension $|\mathcal{H}_{k,t}|$.

Besides Reboot [17], which has been analyzed only for Gaussian distributions, all other algorithms have been analyzed for distributions with known bounded support, for which they are proved to have logarithmic regret. Among them, Non Parametric TS has strong optimality property as its regret is proved to match the lower bound of Burnetas and Katehakis [19] for (non-parametric) distribution that are bounded in [0,1]. In this paper, we propose the first re-sampling based algorithm that is asymptotically optimal for several classes of possibly un-bounded parametric distributions. We introduce a new family of algorithms called Sub-Sampling Duelling Algorithms, and provide a regret analysis for RB-SDA, an algorithm based on *Random Block* sub-sampling. In Theorem 3.1, we show that RB-SDA has logarithmic regret under some general conditions on the arms distributions. These conditions are in particular satisfied for Gaussian, Bernoulli and Poisson distribution, for which we further prove in Corollary 3.1.1 that RB-SDA is asymptotically optimal.

The general SDA framework that we introduce is inspired by two ideas first developed for the BESA algorithm by [1] and for the SSMC algorithm by [2]: 1) the arms pulled are chosen according to the outcome of pairwise comparison (*duels*) between arms, instead of choosing the maximum of some index computed for each arm as GRE algorithms do, and 2) the use of *sub-sampling*: the algorithm penalizes arms that have been pulled a lot by making them compete with the other arms with only a fraction of their history. More precisely, in a duel between two arms $A$ and $B$ selected $n_A$ and $n_B$ times respectively, with $n_A < n_B$, the empirical mean of arm $A$ is compared to the empirical mean of a sub-sample of size $n_A$ of the history of arm $B$. In BESA the duels are organized in a tournament and only the winner is sampled, while SSMC uses rounds of $K-1$ duels between an arm called *leader* and all other arms. Then the leader is pulled only if it wins all the duels, otherwise all the winning *challengers* are pulled. Second difference is that in BESA the sub-sample of the leader's history is obtained with *Sampling Without Replacement*, whereas SSMC selects this sub-sample as the block of consecutive observations with smallest empirical mean. Hence BESA uses randomization while SSMC does not. Finally, SSMC also uses some forced exploration (i.e. selects any arm drawn less than $\sqrt{\log r}$ times in round $r$). In SDA, we propose to combine the round structure for the duels used by SSMC with the use of a sub-sampling scheme assumed to be independent of the observations in the history (this generalizes the BESA duels), and we get rid of the use of forced exploration.

The rest of the paper is structured as follows. In Section 2 we introduce the SDA framework and present different instances that correspond to the choice of different sub-sampling algorithms, in particular RB-SDA. In Section 3 we present upper bounds on the regret of RB-SDA, showing in particular that the algorithm is asymptotically optimal for different exponential families. We sketch the proof of Theorem 3.1 in Section 4, highlighting two important tools: First, a new concentration lemma for random sub-samples (Lemma 4.2). Second, an upper bound on the probability that the optimal arm is under-sampled, which decouples the properties of the sub-sampling algorithm used, and that of the arms' distributions (Lemma 4.3). Finally, Section 5 presents the results of an empirical study comparing several instances of SDA to asymptotically optimal parametric algorithms and other

algorithms based on re-sampling or sub-sampling. These experiments reveal the robustness of the SDA approaches, which match the performance of Thompson Sampling, without exploiting the knowledge of the distribution.

## 2 Sub-sampling Duelling Algorithms

In this section, we introduce the notion of Sub-sampling Duelling Algorithm (SDA). We first introduce a few notation. For every integer $n$, we let $[n] = \{1, \ldots, n\}$. We denote by $(Y_{k,s})_{s \in \mathbb{N}}$ the i.i.d. sequence of successive rewards from arm $k$, that are i.i.d. under a distribution $\nu_k$ with mean $\mu_k$. For every finite subset $\mathcal{S}$ of $\mathbb{N}$, we denote by $\hat{Y}_{k,\mathcal{S}}$ the empirical mean of the observations of arm $k$ indexed by $\mathcal{S}$: if $|\mathcal{S}| > 1$, $\hat{Y}_{k,\mathcal{S}} := \frac{1}{|\mathcal{S}|} \sum_{i \in \mathcal{S}} Y_{k,i}$. We also let $\hat{Y}_{k,n}$ as a shorthand notation for $\hat{Y}_{k,[n]}$.

**A round-based algorithm** Unlike index policies, a SDA algorithm relies on *rounds*, in which several arms can be played (at most once). In each round $r$ the learner selects a subset of arms $\mathcal{A}_r = \{k_1, ..., k_{i_r}\} \subseteq \{1, \ldots, K\}$, and receives the rewards $\mathcal{X}_r = \{Y_{k_1, N_{k_1}(r)}, ..., Y_{k_{i_r}, N_{k_{i_r}}(r)}\}$ associated to the chosen arms, where $N_k(r) := \sum_{s=1}^{r} \mathbb{1}(k \in \mathcal{A}_s)$ denotes the number of times arm $k$ was selected up to round $r$. Letting $\hat{r}_T \leq T$ be the (random) number of rounds used by algorithm $\mathcal{A}$ before the $T$-th arm selection, the regret of a round-based algorithm can be upper bounded as follows:

$$
\begin{aligned}
\mathcal{R}_T(\mathcal{A}) = \mathbb{E}\left[\sum_{t=1}^{T}(\mu_\star - \mu_{A_t})\right] &\leq \mathbb{E}\left[\sum_{s=1}^{\hat{r}_T}\sum_{k=1}^{K}(\mu_\star - \mu_k)\mathbb{1}(k \in \mathcal{A}_s)\right] \\
&\leq \mathbb{E}\left[\sum_{s=1}^{T}\sum_{k=1}^{K}(\mu_\star - \mu_k)\mathbb{1}(k \in \mathcal{A}_s)\right] = \sum_{k=1}^{K}(\mu_\star - \mu_k)\mathbb{E}\left[N_k(T)\right] . \quad (2)
\end{aligned}
$$

Hence upper bounding $\mathbb{E}[N_k(T)]$ for each sub-optimal arm provides a regret upper bound.

**Sub-sampling Duelling Algorithms** A SDA algorithm takes as input a *sub-sampling algorithm* $\mathrm{SP}(m, n, r)$ that depends on three parameters: two integers $m \geq n$ and a round $r$. A call to $\mathrm{SP}(m, n, r)$ at round $r$ produces a subset of $[m]$ that has size $n$, modeled as a random variable that is further assumed to be independent of the rewards generated from the arms, $(Y_{k,s})_{k \in [K], s \in \mathbb{N}^*}$.

In the first round, a SDA algorithm selects $\mathcal{A}_1 = [K]$ in order to initialize the history of all arms. For $r \geq 1$, at round $r + 1$, a SDA algorithm based on a sampler SP, that we refer to as SP-SDA, first computes the *leader*, defined as the arm being selected the most in the first $r$ round: $\ell(r) = \mathrm{argmax}_k N_k(r)$. Ties are broken in favor of the arm with the largest mean, and if several arms share this mean then the previous leader is kept or one of these arms is chosen randomly. Then the set $\mathcal{A}_{r+1}$ is initialized to the empty set and $K - 1$ *duels* are performed. For each "challenger" arm $k \neq \ell(r)$, a subset $\hat{S}_k^r$ of $[N_{\ell(r)}(r)]$ of size $N_k(r)$ is obtained from $\mathrm{SP}(N_{\ell(r)}(r), N_k(r), r)$ and arm $k$ wins the duels if its empirical mean is larger than the empirical mean of the sub-sampled history of the leader. That is

$$
\hat{Y}_{k, N_k(r)} > \hat{Y}_{\ell(r), \hat{S}_k^r} \implies \mathcal{A}_{r+1} = \mathcal{A}_{r+1} \cup \{k\} .
$$

If the leader wins all the duels, that is if $\mathcal{A}_{r+1}$ is still empty after the $K - 1$ duels, we set $\mathcal{A}_{r+1} = \{\ell(r)\}$. Arms in $\mathcal{A}_{r+1}$ are then selected by the learner in a random order and are pulled if the total budget of pulls remains smaller than $T$. The pseudo-code of SP-SDA is given in Algorithm 1.

To properly define the random variable $\hat{S}_k^r$ used in the algorithm, we introduce the following probabilistic modeling: for each round $r$, each arm $k$, we define a family $(S_k^r(m, n))_{m \geq n}$ of independent random variables such that $S_k^r(m, n) \sim \mathrm{SP}(m, n, r)$. In words, $S_k^r(m, n)$ is the subset of the leader history used should arm $k$ be a challenger drawn $n$ times up to round $r$ dueling against a leader that has been drawn $m$ times. With this notation, for each arm $k \neq \ell(r)$ one has $\hat{S}_k^r = S_k^r\left(N_{\ell(r)}(r), N_k(r), r\right)$. We recall that in the SDA framework, it is crucial that those random variables are independent from the reward streams $(Y_{k,s})$ of all arms $k$. We call such sub-sampling algorithms *independent sampler*.

**Particular instances** We now present a few sub-sampling algorithms that we believe are interesting to use within the SDA framework. Intuitively, these algorithms should ensure enough *diversity* in the output subsets when called in different rounds, so that the leader cannot always look good,

---
**Algorithm 1** SP-SDA
---
**Require:** K arms, horizon T, Sampler SP
  $t \leftarrow K, r \leftarrow 1, \forall k, N_k \leftarrow 1, \mathcal{H}_k \leftarrow \{Y_{k,1}\}$ (Each arm is drawn once)
  **while** $t < T$ **do**
    $r \leftarrow r+1, \mathcal{A} \leftarrow \{\}, \ell \leftarrow \text{leader}(N, \mathcal{H}, \ell)$ (Initialize the round)
    **for** $k \neq \ell \in 1, ..., K$ **do**
      Draw $\hat{S}_k^r \sim \text{SP}(N_\ell, N_k, r)$ (Choice of the sub-sample of $\ell$ used for the duel with $k$)
      **if** $\hat{Y}_{k,N_k} > \hat{Y}_{\ell, \hat{S}_k^r}$ **then**
        $\mathcal{A} \leftarrow \mathcal{A} \cup \{k\}$ (Duel outcome)
      **end if**
    **end for**
    **if** $|\mathcal{A}| = 0$ **then**
      $\mathcal{A} \leftarrow \{\ell\}$
    **end if**
    **if** $|\mathcal{A}| > T - t$ **then**
      $\mathcal{A} \leftarrow \text{choose}(\mathcal{A}, T - t)$ (Randomly selects a number of arm that does not exceed the budget)
    **end if**
    **for** $a \in \mathcal{A}$ **do**
      Pull arm $a$, observe reward $Y_{a, N_a+1}$
      $t \leftarrow t+1, N_a \leftarrow N_a + 1, \mathcal{H}_a \leftarrow \mathcal{H}_a \cup \{Y_{a,N_a}\}$ (Update step)
    **end for**
  **end while**
---

and challengers may win and be explored from time to time. The most intuitive candidates are random samplers like *Sampling Without Replacement* (WR) and *Random Block Sampling* (RB): the first one returns a subset of size $n$ selected uniformly at random in $[m]$, while the second draws an element $n_0$ uniformly at random in $[m - n]$ and returns $\{n_0 + 1, ..., n_0 + n\}$. But we also propose two deterministic sub-sampling: *Last Block* (LB) which returns $\{m - n + 1, ..., m\}$, and *Low Discrepancy Sampling* (LDS) that is similar to RB with the first element $n_0$ of the block at a round $r$ defined as $\lceil u_r(m - n) \rceil$ with $u_r$ a predefined low discrepancy sequence [20] (Halton [21], Sobol [22]). We believe that these last two samplers may ensure enough diversity without the need for random sampling. These four variants of SDA will be compared in Section 5 in terms of empirical efficiency and numerical complexity. For RB-SDA, we provide a regret analysis in the next sections, highlighting what parts may or may not be extended to other sampling algorithms.

**Links with existing algorithms** The BESA algorithm [1] with $K = 2$ coincides with WR-SDA. However beyond $K > 2$, the authors of [1] rather suggest a tournament approach, without giving a regret analysis. WR-SDA can therefore be seen as an alternative generalization of BESA beyond 2 arms, which performs much better than the tournament, as can be seen in Section 5. While the structure of SSDA is close to that of SSMC [2], SSMC is not a SP-SDA algorithm, as its sub-sampling algorithm heavily relies on the rewards, and is therefore not an independent sampler. Indeed, it outputs the set $\mathcal{S} = \{n_0 + 1, ..., n_0 + n\}$ for which $\hat{Y}_{\ell(r), \mathcal{S}}$ is the smallest. The philosophy of SSMC is a bit different than that of SSDA: while the former tries to disadvantage the leader as much as possible, the latter only tries to make the leader use different parts of its history. Our experiments reveal that the SSMC approach seems to lead to a slightly larger regret, due to a bit more exploration in the beginning. Finally, we emphasize that alternative algorithms based on re-sampling (PHE, Reboot, Non-Parametric TS) are fundamentally different to SDA as they do not perform *sub-sampling*.

**On the use of forced exploration** In[2], SSMC additionally requires some *forced exploration*: each arm $k$ such that $N_k(r)$ is smaller than some value $f_r$ is added to $\mathcal{A}_{r+1}$. SSMC is proved to be asymptotically optimal for exponential families provided that $f_r = o(\log r)$ and $\log \log r = o(f_r)$. In the next section, we show that RB-SDA does not need forced exploration to be asymptotically optimal for Bernoulli, Gaussian and Poisson distributions. However, we show in Appendix H that adding forced exploration to RB-SDA is sufficient to prove its optimality for any exponential family.

# 3  Regret Upper Bounds for RB-SDA

In this section, we present upper bounds on the expected number of selections of each sub-optimal arm $k$, $\mathbb{E}\left[N_k(T)\right]$, for the RB-SDA algorithm. They directly yield an upper bound on the regret via (2). To ease the presentation, we assume that there is a unique optimal arm[1], and denote it by 1.

In Theorem 3.1, we first identify some conditions on the arms distribution under which RB-SDA has a regret that is provably logarithmic in $T$. In order to introduce these conditions, we recall the definition of the following *balance function*, first introduced by [1]. $\alpha_k(M, j)$ is equal to the probability that arm 1 loses a certain amount $M$ of successive duels against $M$ sub-samples from arm $k$ that have non-overlapping support, when arm 1 has been sampled $j$ times.

**Definition.** *Letting $\nu_{k,j}$ denote the distribution of the sum of $j$ independent variables drawn from $\nu_k$, and $F_{\nu_{k,j}}$ its corresponding CDF, the balance function of arm $k$ is*

$$\alpha_k(M, j) = \mathbb{E}_{X \sim \nu_{1,j}} \left( \left( 1 - F_{\nu_{k,j}}(X) \right)^M \right) .$$

**Theorem 3.1** (Logarithmic Regret for RB-SDA). *If the arms distributions $\nu_1, \ldots, \nu_k$ are such that*

1. *the empirical mean of each arm $k$ has exponential concentration given by a certain rate function $I_k(x)$ which is continuous and satisfies $I_k(x) = 0$ if and only if $x = \mu_k$:*

$$\forall x > \mu_k, \mathbb{P}\left( \hat{Y}_{k,n} \geq x \right) \leq e^{-n I_k(x)} \text{ and } \forall x < \mu_k, \mathbb{P}\left( \hat{Y}_{k,n} \leq x \right) \leq e^{-n I_k(x)} ,$$

2. *the balance function of each sub-optimal arm $k$ satisfies*

$$\forall \beta \in (0, 1), \ \sum_{t=1}^{T} \sum_{j=1}^{\lfloor (\log t)^2 \rfloor} \alpha_k\left( \lfloor \beta t/(\log t)^2 \rfloor, j \right) = o(\log T) .$$

*Then, for all sub-optimal arm $k$, for all $\varepsilon > 0$, under RB-SDA*

$$\mathbb{E}[N_k(T)] \leq \frac{1 + \varepsilon}{I_k(\mu_1)} \log(T) + o(\log T) .$$

If the distributions belong to the same one-dimensional exponential family (see e.g. [6] for a presentation of some of their important properties), the Chernoff inequality tells us that the concentration condition 1. is satisfied with a rate function equal to $I_k(x) = \mathrm{kl}(x, \mu_k)$ where $\mathrm{kl}(x, y)$ is the Kullback-Leibler divergence between the distribution of mean $x$ and the distribution of mean $y$ in that exponential family. In Appendix G, we prove that Gaussian distribution with known variance, Bernoulli and Poisson distribution also satisfy the balance condition 2., which yields the following.

**Corollary 3.1.1.** *Assume that the distribution of all arms belong to the family of Gaussian distributions with a known variance, Bernoulli or Poisson distributions. Then under RB-SDA for all $\varepsilon > 0$, for all sub-optimal arm $k$,*

$$\mathbb{E}[N_k(T)] \leq \frac{1 + \varepsilon}{\mathrm{kl}(\mu_k, \mu_1)} \log(T) + o_{\varepsilon, \boldsymbol{\mu}}(\log(T)).$$

Corollary 3.1.1 permits to prove that $\limsup_{T \to} \frac{\mathcal{R}_T(\text{RB-SDA})}{\log(T)} \leq \sum_{k=2}^{K} \frac{(\mu_1 - \mu_k)}{\mathrm{kl}(\mu_k, \mu_1)}$, which is matching the Lai & Robbins lower bound (1) in each of these exponential families. In particular, RB-SDA is simultaneously asymptotically optimal for different examples of bounded (Bernoulli) and un-bounded (Poisson, Gaussian) distributions. In contrast, Non-Parametric TS is asymptotically optimal for any bounded distributions, but cannot be used for Gaussian or Poisson distributions. Note that the guarantees of Corollary 3.1.1 also hold for the SSMC algorithm [2], but we prove that RB-SDA can be asymptotically optimal *without forced exploration* for some distributions. Moreover, as will be seen in Section 5, algorithms based on randomized history-independent sub-sampling such as RB-SDA tend to perform better than deterministic algorithms such as SSMC.

Theorem 3.1 also shows that RB-SDA may have logarithmic regret for a wider range of distributions. For example, we conjecture that a truncated Gaussian distribution also satisfy the balance condition

2.. On the other hand, condition 2. does not hold for Exponential distribution, as discussed in Appendix G.4. But we show in Appendix H.2 that any distribution that belongs to a one-dimensional exponential family satisfies a slightly modified version of this condition, which permits to establish the asymptotic optimality of a variant of RB-SDA using forced exploration.

Finally, we note that it is possible to build on RB-SDA to propose a bandit algorithm that has logarithmic regret for any distribution that is bounded in $[0, 1]$. To do so, we can use the binarization trick already proposed by [11] for Thompson Sampling, and run RB-SDA on top of a binarized history $\mathcal{H}'_k$ for each arm $k$ in which a reward $Y_{k,s}$ is replaced by a binary pseudo-reward is $Y'_{k,s}$ generated from a Bernoulli distribution with mean $Y_{k,s}$. The resulting algorithm inherits the regret guarantees of RB-SDA applied to Bernoulli distributions.

Characterizing the set of distributions for which the vanilla RB-SDA algorithm has logarithmic regret (without forced exploration or a binarization trick) is left as an interesting future work.

# 4 Sketch of Proof

In this section, we provide elements of proof for Theorem 3.1, postponing the proof of some lemmas to the appendix. The first step is given by the following lemma, which is proved in Appendix D.

**Lemma 4.1.** *Under condition 1., for any SP-SSDA algorithm (using an independent sampler), for every $\varepsilon > 0$, there exists a constant $C_k(\boldsymbol{\nu}, \varepsilon)$ with $\boldsymbol{\nu} = (\nu_1, \ldots, \nu_k)$ such that*

$$\mathbb{E}[N_k(T)] \leq \frac{1+\varepsilon}{I_1(\mu_k)} \log(T) + 32 \sum_{r=1}^{T} \mathbb{P}\left(N_1(r) \leq (\log(r))^2\right) + C_k(\boldsymbol{\nu}, \varepsilon) \ .$$

The proof of this result follows essentially the same decomposition as the one proposed by [2] for the analysis of SSMC. However, it departs from this analysis in two significant ways. First, instead of using properties of forced exploration (that is absent in RB-SDA), we distinguish whether or not arm 1 has been selected a lot, which yields the middle term in the upper bound. Then, the argument relies on a new concentration result for sub-samples averages, that we state below. Lemma 4.2, proved in Appendix C, crucially exploits the fact that a SP-SDA algorithm is based on an independent sampler. Using condition 1. allows to further upper bound the right-hand side of the two inequalities in Lemma 4.2 by terms that decay exponentially and contribute to the constant $C_k(\boldsymbol{\nu}, \varepsilon)$.

**Lemma 4.2** (concentration of a sub-sample). *For all $(a, b)$ such that $\mu_a < \mu_b$, for all $\xi \in (\mu_a, \mu_b)$ and $n_0 \in \mathbb{N}$, under any instance of SP-SDA using an independent sampler, it holds that*

$$\sum_{s=1}^{r} \mathbb{P}\left(\hat{Y}_{a,N_a(s)} \geq \hat{Y}_{b,\hat{S}_b^s(N_b(s),N_a(s))}, N_b(s) \geq N_a(s), N_a(s) \geq n_0\right) \leq \sum_{j=n_0}^{r} \mathbb{P}\left(\hat{Y}_{a,j} \geq \xi\right) + r \sum_{j=n_0}^{r} \mathbb{P}(Y_{b,j} \leq \xi),$$

$$\sum_{s=1}^{r} \mathbb{P}\left(\hat{Y}_{b,N_b(s)} \leq \hat{Y}_{a,\hat{S}_a^s(N_a(s),N_b(s))}, N_a(s) \geq N_b(s), N_b(s) \geq n_0\right) \leq \sum_{j=n_0}^{r} \mathbb{P}\left(\hat{Y}_{b,j} \leq \xi\right) + r \sum_{j=n_0}^{r} \mathbb{P}(Y_{a,j} \geq \xi).$$

So far, we note that the analysis has *not* been specific to RB-SDA but applies to any instance of SDA. Then, we provide in Lemma 4.3 an upper bound on $\sum_{t=1}^{T} \mathbb{P}\left(N_1(t) \leq (\log t)^2\right)$ which is specific to RB-SDA. This sampler is randomized and independent of $r$, hence we use the notation $\mathrm{RB}(m, n) = \mathrm{RB}(m, n, r)$. The strength of this upper bound is that it decouples the properties of the sub-sampling algorithm and that of the arm distributions (through the balance function $\alpha_k$).

**Lemma 4.3.** *Let $X_{m,H,j}$ be a random variable giving the number of non-overlapping sub-samples of size $j$ obtained in $m$ i.i.d. samples from $\mathrm{RB}(H, j)$ and define $c_r = \lfloor \frac{r/(\log r)^2 - 1}{2K} \rfloor - 1$. There exists $\gamma \in (0, 1)$ and a constant $r_K$ such that with $\beta_{r,j} = \lfloor \gamma r/j(\log r)^2 \rfloor$,*

$$\sum_{r=1}^{T} \mathbb{P}(N_1(r) \leq (\log r)^2) \leq r_K + \sum_{r=r_K}^{T} \sum_{j=1}^{\lfloor \log r^2 \rfloor} \left[(K-1)\mathbb{P}\left(X_{c_r,c_r,j} < \beta_{r,j}\right) + \sum_{k=2}^{K} \alpha_k\left(\beta_{r,j}, j\right)\right] \ .$$

To prove Lemma 4.3 (see Appendix E), we extend the proof technique introduced by [1] for the analysis of BESA to handle more than 2 arms. The rationale is that if $N_1(r) \leq (\log r)^2$ then arm 1 is not the leader and has lost "many" duels, more precisely *at least* a number of *successive duels*

proportional to $r/\left(\log r\right)^2$. A fraction of these duels necessarily involves sub-samples of the leader history that have non-overlapping support. Exploiting the independence of these sub-samples brings in the balance function $\alpha_k$.

In order to conclude the proof, it remains to upper bound the right hand side of Lemma 4.3. Using condition 2. of balanced distributions the terms depending on $\alpha_k$ sum in $o(\log T)$ and negligibly contribute to the regret. Upper bounding the term featuring $X_{m,H,j}$ amounts to establishing the following diversity property of the random block sampler.

**Definition** (Diversity Property). *Let $X_{m,H,j}$ be the random variable defined in Lemma 4.3 for a randomized sampler* SP. SP *satisfies the Diversity Property for a sequence $N_r$ of integers if*

$$\sum_{r=1}^{T} \sum_{j=1}^{(\log r)^2} \mathbb{P}\left(X_{N_r,N_r,j} < \gamma r/(\log r)^2\right) = o(\log T).$$

We prove in Appendix F that the RB sampler satisfies the diversity property for the sequence $c_r$, which leads to $\sum_{t=1}^{T} \mathbb{P}\left(N_1(t) \le (\log t)^2\right) = o(\log(T))$ and concludes the proof of Theorem 3.1.

We believe that the WR sampler also satisfies the diversity property (as conjectured by [1]). While Lemma 4.3 should apply to WR-SDA as well, a different path has to be found for analyzing the LDS-SDA and LB-SDA algorithms, that are based on deterministic samplers and also perform well in practice. This is left for future work.

## 5   Experiments

In this section, we perform experiments on simulated data in order to illustrate the good performance of the four instances of SDA algorithms introduced in Section 2 for various distributions. The Python code used to perform these experiments is available on Github.

**Exponential families**   First, in order to illustrate Corollary 3.1.1, we investigate the performance of RB-SDA for both Bernoulli and Gaussian distributions (with known variance 1). Our first objective is to check that for a finite horizon the regret of RB-SDA is comparable with the regret of Thompson Sampling (with respectively a beta and improper uniform prior), which efficiently uses the knowledge of the distribution. Our second objective is to empirically compare different variants of SDA to other non-parametric approaches based on sub-sampling (BESA, SSMC) or on re-sampling. For Bernoulli and Gaussian distribution, Non-Parameteric TS coincides with Thompson Sampling, so we focus our study on algorithms based on history perturbation. We experiment with PHE [14] for Bernoulli bandits and ReBoot [17] for Gaussian bandits, as those two algorithms are guaranteed to have logarithmic regret in each of these settings. As advised by the authors, we use a parameter $a = 1.1$ for PHE and $\sigma = 1.5$ for ReBoot.

We ran experiments on 4 different Bernoulli bandit models: 1) $K = 2$, $\mu = [0.8, 0.9]$, 2) $K = 2$, $\mu = [0.5, 0.6]$, 3) $K = 10$, $\mu_1 = 0.1, \mu_{2,3,4} = 0.01, \mu_{5,6,7} = 0.03, \mu_{8,9,10} = 0.05$, 4) $K = 8$ $\mu = [0.9, 0.85, \dots, 0.85]$ and 3 different bandits models with $\mathcal{N}(\mu_k, 1)$ arms with means: 1) $K = 2$ $\mu = [0.5, 0]$, 2) $K = 4$, $\mu = [0.5, 0, 0, 0]$, 3) $K = 4$, $\mu = [1.5, 1, 0.5, 0]$. For each experiment, Table 1 and Table 2 report an estimate of the regret at time $T = 20000$ based on 5000 independent runs (extended tables with standard deviations can be found in Appendix A.1). The best performing algorithms are highlighted in bold. In Figure 1 and Figure 2 we plot the regret of several algorithms as a function of time (in log scale) for $t \in [15000; 20000]$ for one Bernoulli and one Gaussian experiment respectively. We also add the Lai and Robbins lower bound $t \mapsto [\sum_k (\mu^\star - \mu_k)/\mathrm{kl}(\mu_k, \mu_\star)] \log(t)$.

Table 1: Regret at $T = 20000$ for Bernoulli arms

| xp | TS | PHE | BESA | SSMC | RB | WR | LB | LDS |
|----|----|----|----|----|----|----|----|----|
| | | Benchmark | | | SDA | | | |
| 1 | **11.2** | 25.9 | **11.7** | 12.3 | **11.5** | **11.6** | 12.2 | **11.4** |
| 2 | **22.9** | 24.0 | **22.1** | 24.3 | **22.0** | **21.5** | 24.0 | **21.8** |
| 3 | **94.2** | 248.1 | **88.1** | 100.1 | **89.0** | **86.9** | 100.7 | **89.2** |
| 4 | **108.1** | 216.5 | 147.5 | 119.9 | **105.1** | **106.9** | 119.6 | **106.8** |

Figure 1: Regret as a function of time for Bernoulli experiment 3

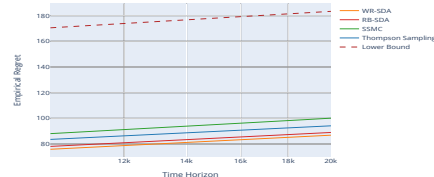

Table 2: Regret at $T = 20000$ for Gaussian arms

| | Benchmark | | | | SDA | | | |
|---|---|---|---|---|---|---|---|---|
| xp | TS | ReBoot | BESA | SSMC | RB | WR | LB | LDS |
| 1 | **24.4** | 92.2 | **25.3** | **26.9** | **25.6** | **24.7** | **25.1** | 26.5 |
| 2 | **73.5** | 277.1 | 122.5 | **74.8** | **71.0** | **71.1** | **74.6** | 69.0 |
| 3 | **49.7** | 190.9 | 72.1 | **51.3** | **50.4** | **50.0** | **51.2** | 48.6 |

Figure 2: Regret as a function of time for Gaussian experiment 2

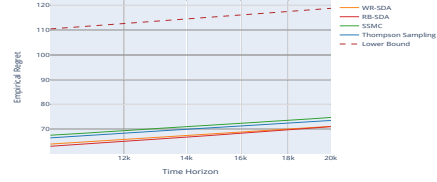

In all of these experiments, we notice that SDA algorithms are indeed strong competitors to Thompson Sampling (with appropriate prior) for both Bernoulli and Gaussian bandits. Figures 1 and 2 further show that RB-SDA is empirically matching the Lai and Robbins' lower bound on two instances, just like SSMC and Thompson Sampling, which can be seen from the parallel straight lines with the $x$ axis in log scale. The fact that the lower bound is above shows that it is really asymptotic and only captures the right first order term. The same observation was made for all experiments, but is not reported due to space limitation. Even if we only established the asymptotic optimality of RB-SDA, these results suggest that the other SDA algorithms considered in this paper may also be asymptotically optimal. Compared to SDA, re-sampling algorithms based on history perturbation seem to be much less robust. Indeed, in the Bernoulli case, PHE performs very well for experiment 2, but is significantly worse than Thompson Sampling on the three other instances. In the Gaussian case, ReBoot always performs significantly worse than other algorithms. This lack of robustness is also corroborated by additional experiments reported in Appendix **??** in which we average the performance of these algorithms over a large number of randomly chosen instances.

Turning our attention to algorithms based on sub-sampling, we first notice that WR-SDA seems to be a better generalization of BESA with 2 arms than the tournament approach currently proposed, as in experiments with $K > 2$, WR-SDA often performs significantly better than BESA. Then we observe that SSMC and SDA algorithms have similar performance. Looking a bit closer, we see that the performance of SSMC is very close to that of LB-SDA, whereas SDA algorithms based on "randomized" (or pseudo-randomized for LDS-SDA) samplers tend to perform slightly better.

**Truncated Gaussian** Theorem 3.1 suggests that RB-SDA may attain logarithmic regret beyond exponential families. As an illustration, we present the results of experiments performed with Truncated Gaussian distributions (in which the distribution of arm $k$ is that of $Y_k = 0 \vee (X_k \wedge 1)$ where $X_k \sim \mathcal{N}(\mu_k, \sigma^2)$). We report in Table 8 the regret at time $T = 20000$ (estimated over 5000 runs) of various algorithms on four different problem instances: 1) $\mu = [0.5, 0.6]$, $\sigma = 0.1$, 2) $\mu = [0, 0.2]$, $\sigma = 0.3$, 3) $\mu = [1.5, 2]$, $\sigma = 1$ 4) $\mu = [0.4, 0.5, 0.6, 0.7]$, $\sigma = 1$. We include Non-Parametric TS which is known to be asymptotically optimal in this setting (while TS which uses a Beta prior and a binarization trick is not), PHE, and all algorithms based on sub-sampling. We again observe the good performance of SSMC and SDA algorithms across all experiments. They even outperform NP-TS in some experiments, which suggests SDA algorithms may be asymptotically optimal for a wider class of parametric distributions.

Table 3: Regret at $T = 20000$ for Truncated Gaussian arms

| | Benchmark | | | | | SDA | | | |
|---|---|---|---|---|---|---|---|---|---|
| xp | TS | NP-TS | PHE | BESA | SSMC | RB | WR | LB | LDS |
| 1 | 21.9 | 4.2 | 22.3 | **1.4** | **1.5** | **1.4** | **1.4** | **1.5** | **1.4** |
| 2 | 13.3 | 8 | 19.5 | **4.6** | **4.7** | **4.4** | **4.5** | **4.6** | **4.3** |
| 3 | 9.7 | **7.8** | 48.5 | **7.8** | 7.6 | 7.1 | 7.7 | 8.2 | 7.1 |
| 4 | 86.6 | **70** | 86 | 76.5 | **69.5** | **64.9** | **64.8** | **68.7** | 63.2 |

**Bayesian Experiments** So far we tried our algorithms on specific instances of the distributions we considered. It is also interesting to check the robustness of the algorithms when the means of the arms are drawn at random according to some distribution. In this section we consider two examples:

Bernoulli bandits where the arms are drawn uniformly at random in $[0, 1]$, and Gaussian distributions with the mean parameter of each arm itself drawn from a gaussian distribution $\mu_k \sim \mathcal{N}(0, 1)$. In both cases we draw 10000 random problems with $K = 10$ arms and run the algorithms for a time horizon $T = 20000$. We experiment with TS, SSMC, RB-SDA and WR-SDA and also add the IMED algorithm ([23]) which is an asymptotically optimal algorithm that uses the knowledge of the distribution. We do not add LDS-SDA and LB-SDA as they are similar to RB-SDA and SSMC, respectively. In the Bernoulli case, we also run the PHE algorithm, which fails to compete with the other algorithms. This is not in contradiction with the results of [14] as in the Bayesian experiments of this paper, arms are drawn uniformly in $[0.25, 0.75]$ instead of $[0, 1]$. Actually, we noticed that PHE with parameter $a = 1.1$ has some difficulties when several arms are close to 1.

Table 4: Average Regret on 10000 random experiments with Bernoulli Arms

| T | TS | IMED | PHE | SSMC | RB | WR |
|---|---|---|---|---|---|---|
| 100 | 13.8 | 15.1 | 16.7 | 16.5 | 14.8 | 14.3 |
| 1000 | 27.8 | 31.9 | 39.5 | 34.2 | 31.8 | 30.9 |
| 10000 | 45.8 | 51.2 | 72.3 | 55.0 | 51.1 | 50.6 |
| 20000 | 52.2 | 57.6 | 85.6 | 61.9 | 57.7 | 57.3 |

Table 5: Average Regret on 10000 random experiments with Gaussian Arms

| T | TS | IMED | WR | RB | SSMC |
|---|---|---|---|---|---|
| 100 | 41.2 | 45.1 | 38.3 | 38.1 | 40.6 |
| 1000 | 76.4 | 82.1 | 72.7 | 70.4 | 76.2 |
| 10000 | 118.5 | 124.0 | 115.8 | 111.8 | 120.1 |
| 20000 | 132.6 | 138.1 | 130.2 | 125.7 | 135.1 |

Results reported in Tables 4 and 5 show that RB-SDA and WR-SDA are strong competitors to TS and IMED for both Bernoulli and Gaussian bandits. Recall that these algorithm operate without the need for a specific tuning for each distribution, unlike TS and IMED. Moreover, observe that in the Bernoulli case, TS further uses the same prior as that from which the means are drawn.

**Computational aspects** To choose a sub-sampling based algorithm, numerical consideration can be taken into account. First, compared to Thompson Sampling, all sub-sampling based algorithm require to store the history of the observation. But then, the cost of sub-sampling varies across algorithms: in the general case RB-SDA is more efficient than WR-SDA as the latter requires to draw a random subset while the former only needs to draw the random integer starting the block. However, for distributions with finite supports WR-SDA can be made as efficient as TS using multivariate geometric distributions, just like PHE does. If one does not want to use randomization then LDS-SDA could be preferred to RB-SDA as it uses a deterministic sequence. Finally, LB-SDA has the smallest computational cost in the SDA family and while its performance is very close to that of SSMC, it can avoid the cost of scanning all the sub-sample means in this algorithm. The computational cost of these two algorithms is difficult to evaluate precisely. Indeed, they can be made very efficient when the leader does not change, but each change of leader is costly, in particular for SSMC. The expected number of such changes is proved to be finite, but for experiments with a finite time horizon the resulting constant can be big. Finally, Non-Parametric TS has a good performance for Truncated Gaussian, but the cost of drawing a random probability vector over a large history is very high.

**More experiments** In Appendix A we enhance this empirical study: we show some limitations of SDA for exponential distributions and propose a fix using forced exploration as in SSMC.

## 6 Conclusion

We introduced the SDA framework for exploration in bandits models. We proved that one particular instance, RB-SDA, combines both optimal theoretical guarantees and good empirical performance for several distributions, possibly with unbounded support. Moreover, SDA can be associated with other samplers that seem to achieve similar performance, with their own specificity in terms of computation time. The empirical study presented in the paper also shows the robustness of *sub-sampling* approach over other types of *re-sampling* algorithms. This new approach to exploration may be generalized in many directions, for example to contextual bandits or reinforcement learning, where UCB and Thompson Sampling are still the dominant approaches. It is also particularly promising to develop new algorithm for non-stationary bandit, as such algorithms already store the full history of rewards.

## Broader Impact

This work is advertising a new way to do non-parametric exploration in bandit models, that enjoy good empirical performance and strong theoretical guarantees. First, bandit problems are at the heart of numerous applications to online content recommendation, hence the good performance of SDA algorithms may inspire new algorithms for more realistic models used for these applications, such as contextual bandits. Then, exploration is a central question in the broader field of reinforcement learning, hence new ideas for bandits may lead to new ideas for reinforcement learning.

## Acknowledgments and Disclosure of Funding

The PhD of Dorian Baudry is funded by a CNRS80 grant. The authors acknowledge the funding of the French National Research Agency under projects BADASS (ANR-16-CE40-0002) and BOLD (ANR-19-CE23-0026-04).

Experiments presented in this paper were carried out using the Grid'5000 testbed, supported by a scientific interest group hosted by Inria and including CNRS, RENATER and several Universities as well as other organizations (see https://www.grid5000.fr).

## Footnotes

[1]as can be seen in the analysis of SSMC [2], treating the general case only requires some additional notation.

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
