[Supplementary Material]

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

# A Complement of Experiments

## A.1 Additional Figures for Bernoulli, Gaussian and Truncated Gaussian arms

We enhance the tables of Section 5 with the standard deviation (reported in parenthesis) of the regret at time $T = 20000$ on the 5000 trajectories.

Table 6: Regret and at $T = 20000$ for Bernoulli arms, with standard deviation

| xp | Benchmark | | | | SSDA | | | |
|---|---|---|---|---|---|---|---|---|
| | TS | PHE | BESA | SSMC | RB | WR | LB | LDS |
| 1 | **11.2** (10.) | 25.9 (87.9) | **11.7** (12.1) | **12.3** (7.3) | **11.5** (10.1) | **11.6** (10.2) | **12.2** (7.4) | **11.4** (9.0) |
| 2 | **22.9** (29.2) | **24.0** (22.0) | **22.1** (25.2) | **24.3** (38.2) | **22.0** (34.5) | **21.5** (17.3) | **24.0** (24.6) | **21.8** (24.5) |
| 3 | **94.2** (15.8) | 248.1 (25.5) | **88.1** (89.2) | 100.1 (20.0) | **89.0** (19.8) | **86.9** (21.7) | 100.7 (21.3) | **89.2** (21.8) |
| 4 | **108.1** (45.1) | 216.5 (89.8) | 147.5 (209.8) | 119.9 (40.8) | **105.1** (41.1) | **106.9** (42.1) | 119.6 (42.7) | **106.8** (47.7) |

Table 7: Regret and at $T = 20000$ for Gaussian arms, with standard deviation

| xp | Benchmark | | | | SDA | | | |
|---|---|---|---|---|---|---|---|---|
| | TS | ReBoot | BESA | SSMC | RB | WR | LB | LDS |
| 1 | **24.4** (17.1) | 92.2 (23.4) | **25.3** (27.1) | **26.9** (52.8) | **25.6** (62.8) | **24.7** (20.6) | **25.1** (17.9) | **26.5** (140.2) |
| 2 | **73.5** (107.8) | 277.1 (41.3) | 122.5 (585.5) | **74.8** (34.7) | **71.0** (152.2) | **71.1** (50.2) | **74.6** (35.1) | **69.0** (50.4) |
| 3 | **49.7** (26.9) | 190.9 (29.6) | 72.1 (410.3) | **51.3** (23.7) | **50.4** (156.5) | **50.0** (33.3) | **51.2** (22.4) | **48.6** (41.6) |

Table 8: Regret at $T = 20000$ for Truncated Gaussian arms

| xp | Benchmark | | | | | SDA | | | |
|---|---|---|---|---|---|---|---|---|---|
| | TS | NP-TS | PHE | BESA | SSMC | RB | WR | LB | LDS |
| 1 | 21.9 (20.4) | 4.2 (0.6) | 22.3 (2.6) | **1.4** (1.7) | **1.5** (0.7) | **1.4** (1.1) | **1.4** (0.8) | **1.5** (0.7) | **1.4** (0.8) |
| 2 | 13.3 (7) | 8 (1.8) | 19.5 (3.8) | **4.6** (3.3) | **4.7** (2.3) | **4.4** (4.6) | **4.5** (3.1) | **4.6** (2.4) | **4.3** (2.9) |
| 3 | 9.7 (10.1) | **7.8** (4.5) | 48.5 (217.8) | **7.8** (9.4) | **7.6** (5) | **7.1** (10) | **7.7** (13.4) | 8.2 (27.5) | **7.1** (5.8) |
| 4 | 86.6 (57.8) | **70** (39.4) | 86 (53.7) | 76.5 (113.9) | **69.5** (40.9) | **64.9** (60.5) | **64.8** (43.9) | **68.7** (39.1) | **63.2** (51.1) |

For Bernoulli arms and Truncated Gaussian, the standard deviations of SDA are very similar to that of Thompson Sampling, while the trajectories of PHE and BESA have much more variance in experiment 1 and 4, and on experiments 3 and 4 respectively. For Gaussian arms we remark the low variability of ReBoot, but at the cost of a non-competitive regret. SDA are less homogeneous in this

case: some algorithms have large variance for some instances (LDS-SDA on experiment 1, RB-SDA on experiments 2 and 3). Note that TS also has a high variability in experiment 2.

We believe that this is due to the nature of the Gaussian distribution, and in particular to its balance function: in Appendix G we prove that $\alpha_k(M, j)$ does satisfy Assumption 2. of Theorem 3.1, however the upper bound derived for $\alpha_k(M, j)$ is much larger than the one for Bernoulli distribution, which justifies that "bad runs" in which a good arm looses many duels are more likely to happen in that case, and can explain the larger variance. If one wants to reduce the variance of the regret of SDA we recommend the use of some asymptotically negligible forced exploration, as presented for exponential distribution in Appendix A.2, and for which we prove that the algorithm remains asymptotically optimal in Appendix H.

Finally, as in Section 5, we plot the regret of several algorithms as a function of time (in log scale) for $t \in [10000, 20000]$, this time for the Truncated Gaussian distributions. These plots illustrate the fact that some SDA algorithms may achieve asymptotic optimality for this distribution too, even if it does not belong to a one-parameter exponential family. Indeed, the rate of the regret of all SDA seem too match both the rate of the regret of Non-Parametric TS, which is optimal for this family, and the Burnetas and Katehakis lower bound whose expression is $\left( \sum_{k \neq k^\star} \frac{\mathbb{E}_{X \sim \nu_{k^*}}[X] - \mathbb{E}_{X \sim \nu_k}[X]}{\mathrm{KL}(\nu_k, \nu_{k^*})} \right) \log(T)$ in this particular case, with $\mathrm{KL}(\nu_k, \nu_{k^*}) = p_{0,k} \log\left( \frac{p_{0,k}}{p_{0,*}} \right) + (1 - p_{1,k}) \log\left( \frac{1 - p_{1,k}}{1 - p_{1,*}} \right) + \int_0^1 f_k(x) \log\left( \frac{f_k(x)}{f_*(x)} \right) dx$. $p_{x,k}$ is the value of the CDF of the underlying Gaussian random variable associated with $\nu_k$ in $x$, and $f_k(x)$ the density of this variable in $x$.

Figure 3: SDA vs NP-TS on TG expe 2

Figure 4: SDA vs NP-TS on TG expe 4

## A.2 Experiments with Exponential Arms

In Appendix G, we prove that exponential distributions are *not* balanced (i.e. do not satisfy Assumption 2. of Theorem 3.1), so our theoretical results on the regret of RB-SDA do not apply. However, it is still interesting to test our algorithms for these distributions in order to see if it still achieves a good performance. We performed 6 experiments, with the following mean parameters: 1) $\mu = [1.5, 1]$, 2) $\mu = [0.2, 0.1]$, 3) $\mu = [11, 10]$, 4) $\mu = [4, 3, 2, 1]$, 5) $\mu = [0.4, 0.3, 0.2, 0.1]$, 6) $\mu = [5, 4, 4, 4]$. It is interesting to remark that the standard deviation of an exponential distribution is equal to its mean, so with similar gaps problems are harder when the means are high.

Table 9: Average Regret with Exponential Arms (with std)

| xp | TS | IMED | BESA | SSMC | RB | WR | LB | LDS |
|---|---|---|---|---|---|---|---|---|
| 1 | 48.2 (191.8) | **40.0** (78.4) | 45.7 (114.1) | **41.9** (84.2) | 44.8 (121.4) | 45.4 (134.4) | 46.6 (176.8) | 45.5 (109.7) |
| 2 | 3.8 (9.9) | **3.4** (3.6) | 4.2 (25.1) | **3.6** (41.9) | 4.1 (14.3) | 3.9 (13.4) | 3.9 (8.7) | 5.4 (49.5) |
| 3 | 832.8 (1065.1) | **779.9** (896.9) | 820.5 (1304.6) | 856.9 (1111.0) | 848.4 (1533.3) | **778.4** (1118.7) | 846.7 (1150.1) | 877.7 (1708.7) |
| 4 | 258.3 (519.6) | **234.6** (126.6) | 525.4 (2115.1) | **251.3** (328.3) | 272.6 (692.2) | 262.1 (524.4) | 263.8 (477.9) | 258.4 (599.0) |
| 5 | **25.6** (51.2) | **24.0** (33.6) | 55.7 (219.9) | **25.6** (23.6) | **25.5** (46.7) | **25.0** (24.0) | 26.5 (36.8) | **24.7** (37.6) |
| 6 | **618.7** (672.3) | **603.6** (576.8) | 1184.2 (3096.4) | **627.9** (755.6) | **595.7** (790.7) | **616.0** (780.2) | 652.6 (685.3) | **605.9** (871.4) |

First, we notice that the performance of the SDA in terms of the average regret are reasonable, although less impressive than with the other distributions we tested. IMED is almost always the best algorithm in these experiments, and SSMC performs pretty well on many examples (which is not surprising as SSMC is proved to be asymptotically optimal for exponential distributions). We remark that there is much more variability in the results of RB-SDA, WR-SDA and LDS-SDA than before, where they performed quite similarly. For instance, we notice that on example 3, LDS-SDA and RB-SDA are much worse than WR-SDA. A look at the quantile table for this experiment, which displays the empirical quantiles of $R_T$ estimated over 5000 runs, shows that this is due to a small number of "bad" trajectories for these algorithms:

Table 10: Quantile Table for Experiment 3 with Exponential Arms

| % of runs | TS | IMED | SSMC | RB | WR | LB | LDS |
|---|---|---|---|---|---|---|---|
| 20% | 319.8 | 336.0 | 335.0 | 261.0 | 290.0 | 326.0 | 261.8 |
| 50% | 626.0 | 650.0 | 661.0 | 532.0 | 568.5 | 642.0 | 536.0 |
| 80% | 1122.0 | 1080.0 | 1142.0 | 1006.0 | 1019.0 | 1143.2 | 1020.2 |
| 95% | 1924.1 | 1704.0 | 1846.0 | **2199.0** | 1817.2 | 1869.1 | **2134.1** |
| 99% | 4209.4 | 2632.9 | 3536.8 | **6813.1** | 4146.0 | 3762.3 | **7396.7** |

We see that up to the 80% quantile, RB-SDA and LDS-SDA are even significantly better than IMED. This is very different when we look at the 95% and 99% quantiles, which are much greater for our 2 algorithms (even 2.5 times greater for the 99% quantile).

We believe that this very high variability prevents RB-SDA to have a logarithmic regret for exponential arms. Still, the regret is not as bad as being linear, as using the fact that the balance function $\alpha_k(M, j)$ is of order $\exp(-jC)/M$ permits to prove that $\sum_{r=1}^{T} \mathbb{P}(N_1(r) < \log^2(r)) = \mathcal{O}(\log^2(T))$ (which requires to choose a different $\beta_{r,j}$ in Lemma 4.3). But we also found a solution to obtain (asymptotically optimal) linear regret, which consists in adding an asymptotically negligible amount of *forced exploration* as the SSMC algorithm does. This exploration in $o(\log T)$ avoids trajectories where the optimal arm has a very bad first observation and is not drawn for a very long time. In Appendix H, we prove the asymptotic optimality of RB-SDA with forced exploration $f_r = \sqrt{\log r}$ for any one-dimensional exponential family. In practice, adding this amount of forced exploration to SDA algorithms leads to the following results:

Table 11: Average Regret with Exponential Arms: SDA with forced exploration

| xp | RB | WR | LB | LDS |
|---|---|---|---|---|
| 1 | 44.9 (167.3) | **42.5** (107.4) | **42.4** (60.5) | 45.0 (176.0) |
| 2 | **3.6** (9.2) | **3.4** (2.2) | 4.0 (27.9) | **3.6** (11.2) |
| 3 | 837.5 (1466.1) | **788.5** (1222.1) | 827.7 (1055.3) | 832.3 (1514.6) |
| 4 | **244.8** (403.3) | **238.9** (250.8) | 251.7 (248.5) | **246.0** (323.4) |
| 5 | **23.6** (33.4) | **25.1** (41.0) | **25.4** (23.4) | **24.9** (42.2) |
| 6 | **578.9** (651.9) | **595.1** (561.3) | 631.2 (446.4) | **577.8** (652.7) |

Hence, adding forced exploration results in a noticeable improvement for SDA algorithms, with RB-SDA, WR-SDA and LDS-SDA becoming competitive with IMED (or even slightly better) on most examples. Observe that LB-SDA has again comparable performance with SSMC with this new feature. This is not surprising as we implemented the SSMC algorithm with the same amount of forced exploration $f_r = \sqrt{\log r}$ .

# B   Notation for the Proof

General notations:

- $K$ number of arms
- $\nu_k$ distribution of the arm $k$, with mean $\mu_k$
- we assume that $\mu_1 = \max_{k \in [K]} \mu_k$ so we call the (unique) optimal arm "arm 1"
- $I_k(x)$ some rate function of the arm $k$, evaluated in $x$. For 1-parameter exponential families this function will always be the KL-divergence between $\nu_k$ and the distribution from the same family with mean $x$.
- $N_k(r)$ number of pull of arm $k$ up to (and including) round $r$.
- $Y_{k,i}$ reward obtained at the i-th pull of arm $k$.
- $\hat{Y}_{k,i}$ mean of the i-th first reward of arm $k$, $\hat{Y}_{k,\mathcal{S}}$ mean of the rewards of $k$ on a subset of indices $\mathcal{S} \subset [N_k(r)]$: $\hat{Y}_{k,\mathcal{S}} = \frac{1}{|\mathcal{S}|} \sum_{s \in \mathcal{S}} Y_{k,s}$. If $|\mathcal{S}| = i$, then $Y_{k,i}$ and $Y_{k,\mathcal{S}}$ have the same distribution.
- $\ell(r)$ leader at round $r + 1$, $\ell(r) = \text{argmax}_{k \in [K]} N_k(r)$.
- $\text{SP}(m, n, r)$ sub-sampling algorithm, or Sampler, which returns a sequence of $n$ unique elements out of $[m]$.
- $(S_k^r(m, n))_{m \geq n}$ a family of independent random variables such that $S_k^r(m, n) \sim \text{SP}(m, n, r)$.
- $\mathcal{A}_r$ set of arms pulled at a round $r$.
- $\mathcal{R}_r$ regret at the *end* of round $r$.

Notations for the regret analysis, part relying on concentration:

- $\mathcal{G}_k^r = \cup_{s=1}^{r-1}\{\ell(s) = 1\} \cap \{k \in \mathcal{A}_{s+1}\} \cap \{N_k(s) \geq (1+\varepsilon)\xi_k \log r\}$
- $\mathcal{H}_k^r = \cup_{s=1}^{r-1}\{\ell(s) = 1\} \cap \{k \in \mathcal{A}_{s+1}\} \cap \{N_k(s) \geq J_k \log r\}$
- $\mathcal{Z}^r = \{\ell(r) \neq 1\}$, the leader used for the duels in round $r + 1$ is sub-optimal

- $\mathcal{D}^r = \{\exists u \in [\lfloor r/4 \rfloor, r]$ such that $\ell(u-1) = 1\}$, the leader has been optimal at least once between $\lfloor r/4 \rfloor$ and $r$

- $\mathcal{B}^u = \{\ell(u) = 1, k \in \mathcal{A}_{u+1}, N_k(u) = N_1(u) - 1$ for some arm $k\}$, the optimal arm is leader in $u$ but loses its duel again some arm $k$, that have been pulled enough to possibly take over the leadership at next round

- $\mathcal{C}^u = \{\exists k \neq 1, N_k(u) \geq N_1(u), \hat{Y}_{k, S_1^u(N_k(u), N_1(u))} \geq \hat{Y}_{1, N_1(u)}\}$, the optimal arm is not the leader and has lost its duel against the sub-optimal leader.

- $\mathcal{L}^r = \sum_{u=\lfloor r/4 \rfloor}^r \mathbb{1}_{\mathcal{C}^u}$

Notations for the regret analysis, control of the number of pulls of the optimal arm:

- $r_j$ round of the j-th play of the optimal arm

- $\tau_j = r_{j+1} - r_j$

- $\mathcal{E}_j^r := \{\tau_j \geq r/\log r^2 - 1\}$

- $\mathcal{M}_{j,r}^1 = \left[ r_j + 1, r_j + \left\lfloor \frac{r/\log r^2 - 1}{2} \right\rfloor \right]$

- $\mathcal{M}_{j,r}^2 = \left[ t_j + \left\lceil \frac{r/\log r^2 - 1}{2} \right\rceil, r_j + \lfloor r/\log r^2 \rfloor - 1 \right]$

- $\mathcal{I}_{j,r}^k = \{s \in \mathcal{M}_{j,r}^2 : \ell(s-1) = k\}$

- $\mathcal{W}_{s,j}^k = \left\{ \left\{ \hat{Y}_{1,j} < \hat{Y}_{k, S_1^s(N_k(s), j)} \right\}, N_k(s) \geq c_{r,K}, N_1(s) = j \right\}$

- $\mathcal{F}_{j,M}^{k,r} = \left\{ \exists i_1, ..., i_M \in I_{j,r}^k : \forall m < m' \in [M], S_1^{i_m}(N_k(i_m), j) \cap S_1^{i_{m'}}(N_k(i_{m'}), j) = \emptyset \right\}$

- CDF: Cumulative Distribution Function, PDF: Probability Density Function and PMF: Probability Mass Function.

## C  Concentration Result: Proof of Lemma 4.2

We first recall the probabilistic model introduced in Section 2: for each round $r$, each arm $k$, we define a family $(S_k^r(n,m))_{n>m}$ of independent random variables such that $S_k^r(n,m) \sim \mathrm{SP}(n,m,r)$. Those random variables are also independent from the reward streams $(Y_{k,s})_{s \geq 0}$ of all arms $k$.

$S_k^r(n,m)$ is the subset of the leader history that is used should arm $k$ be a challenger drawn $m$ times up to round $r$ duelling against a leader that has been drawn $n$ times. With this notation, letting $\ell(r)$ be the leader after $r$ rounds, at round $r + 1$, for all $k \neq \ell(r)$,

$$(k \in \mathcal{A}_{r+1}) \Leftrightarrow \left( \hat{Y}_{k, N_k(r)} > \hat{Y}_{\ell(r), S_k^r(N_{\ell(r)}(r), N_k(r))} \right).$$

Let $k$ be an arm such that $\mu_k < \mu_1$. We denote by $[n_1, n_k]$ the set of subset of $\{1, \ldots, n_1\}$ of size $n_k$. We define an event

$$\mathcal{Q}_k^s = \{N_k(s) \geq n_0, \ell(s) = 1, \hat{Y}_{k, N_k(s)} > \hat{Y}_{\ell(s), S_k^s(N_1(s), N_k(s))}\}.$$

Noting that $\{\hat{Y}_{k, N_k(s)} > \hat{Y}_{\ell(s), S_k^s(N_1(s), N_k(s))}\} \subset \{\hat{Y}_{k, N_k(s)} \geq \xi\} \cup \{\hat{Y}_{\ell(s), S_k^s(N_1(s), N_k(s))} \leq \xi\}$ for all $\xi \in \mathbb{R}$, we can write $\mathcal{Q}_k^s \subset \mathcal{Q}_k^{s,1} \cup \mathcal{Q}_k^{s,2}$ where

$$\mathcal{Q}_k^{s,1} = \{N_k(s) \geq n_0, \ell(s) = 1, \hat{Y}_{k, N_k(s)} > \xi\}$$
$$\text{and } \mathcal{Q}_k^{s,2} = \{N_k(s) \geq n_0, \ell(s) = 1, \hat{Y}_{\ell(s), S_k^s(N_1(s), N_k(s))} \leq \xi\}.$$

This yields $\sum_{s=1}^{r} \mathbb{P}(\mathcal{Q}_k^s) \leq \sum_{s=1}^{r} \mathbb{P}(\mathcal{Q}_k^{r,1}) + \sum_{s=1}^{r} \mathbb{P}(\mathcal{Q}_k^{r,2})$, which will later provide the two terms in the bound of the lemma. The first one does not involve sub-sampling and can be upper bounded as:

$$
\sum_{s=1}^{r} \mathbb{P}(\mathcal{Q}_k^{s,1}) \leq \mathbb{E} \sum_{s=1}^{r} \mathbb{1}(N_k(s) \geq n_0) \mathbb{1}(N_1(s) > N_k(s)) \mathbb{1}\left(\hat{Y}_{k,N_k(s)} \geq \xi\right) \mathbb{1}\left(k \in \mathcal{A}_{s+1}\right)
$$

$$
\leq \mathbb{E} \sum_{s=n_0}^{r} \sum_{n_k=n_0}^{r} \mathbb{1}\left(N_k(s) = n_k, k \in \mathcal{A}_{s+1}\right) \mathbb{1}\left(\hat{Y}_{k,n_k} \geq \xi\right)
$$

$$
\leq \mathbb{E} \sum_{n_k=n_0}^{r} \mathbb{1}\left(\hat{Y}_{k,n_k} \geq \xi\right) \underbrace{\sum_{s=n_0}^{r} \mathbb{1}\left(N_k(s) = n_k, k \in \mathcal{A}_{s+1}\right)}_{\leq 1}
$$

$$
\leq \sum_{n_k=n_0}^{r} \mathbb{P}\left(\hat{Y}_{k,n_k} \geq \xi\right),
$$

where in the last inequality we use that the event $(N_k(s) = n) \cap (k \in \mathcal{A}_{s+1})$ can happen at most once for $s \in \{n_0, \dots, r\}$ (a similar trick was used for example in the analysis of kl-UCB [24]).

Upper bounding the second term $B_r = \sum_{s=1}^{r} \mathbb{P}(\mathcal{Q}_k^{r,2})$ is more intricate as it involves both $N_k(s)$ and $N_1(s)$. With a similar method we get:

$$
B_r \leq \mathbb{E} \sum_{s=n_0}^{r} \sum_{n_k=n_0}^{r} \sum_{n_1=n_k}^{r} \sum_{\mathcal{S} \in [n_1, n_k]} \mathbb{1}\left(N_k(s) = n_k, k \in \mathcal{A}_{s+1}\right) \mathbb{1}\left(N_1(s) = n_1\right) \mathbb{1}\left(S_k^s(n_1, n_k) = \mathcal{S}\right) \mathbb{1}\left(\hat{Y}_{\ell,\mathcal{S}} \leq \xi\right)
$$

$$
\leq \mathbb{E} \sum_{n_k=n_0}^{r} \sum_{n_1=n_k}^{r} \sum_{\mathcal{S} \in [n_1, n_k]} \mathbb{1}\left(\hat{Y}_{\ell,\mathcal{S}} \leq \xi\right) \sum_{s=n_0}^{r} \mathbb{1}\left(N_k(s) = n_k, k \in \mathcal{A}_{s+1}\right) \mathbb{1}\left(S_k^s(n_1, n_k) = \mathcal{S}\right)
$$

$$
= \mathbb{E} \sum_{n_k=n_0}^{r} \sum_{n_1=n_k}^{r} \sum_{\mathcal{S} \in [n_1, n_k]} \mathbb{1}\left(\hat{Y}_{\ell,\mathcal{S}} \leq \xi\right) \sum_{s=n_0}^{r} \mathbb{E}\left[\mathbb{1}\left(N_k(s) = n_k, k \in \mathcal{A}_{s+1}\right) \mathbb{1}\left(S_k^s(n_1, n_k) = \mathcal{S}\right) | \mathcal{F}\right],
$$

where $\mathcal{F}$ is the filtration generated by the reward streams. $N_k(s)$ may have a complicated distribution with respect to this filtration but this is not a problem here. Indeed, $S_k^s(n_1, n_k)$ is by design independent of this filtration, and one can write

$$B_r \leq \mathbb{E} \sum_{n_k=n_0}^{r} \sum_{n_1=n_k}^{r} \sum_{\mathcal{S} \in [n_1, n_k]} \mathbb{1} \left( \hat{Y}_{1,\mathcal{S}} \leq \xi \right) \sum_{s=n_0}^{r} \mathbb{P} \left( S_k^s(n_1, n_k) = \mathcal{S} \right) \mathbb{E} \left[ \mathbb{1} \left( N_k(s) = n_k, k \in \mathcal{A}_{s+1} \right) | \mathcal{F} \right]$$

$$= \mathbb{E} \sum_{n_k=n_0}^{r} \sum_{n_1=n_k}^{r} \sum_{\mathcal{S} \in [n_1, n_k]} \mathbb{1} \left( \hat{Y}_{1,\mathcal{S}} \leq \xi \right) \sum_{s=n_0}^{r} \mathbb{P} \left( S_k^s(n_1, n_k) = \mathcal{S} \right) \mathbb{1} \left( N_k(s) = n_k, k \in \mathcal{A}_{s+1} \right)$$

$$= \sum_{n_k=n_0}^{r} \sum_{n_1=n_k}^{r} \sum_{\mathcal{S} \in [n_1, n_k]} \mathbb{P} \left( \hat{Y}_{1,\mathcal{S}} \leq \xi \right) \sum_{s=n_0}^{r} \mathbb{P} \left( S_k^s(n_1, n_k) = \mathcal{S} \right) \mathbb{E} \left( \mathbb{1} \left( N_k(s) = n_k, k \in \mathcal{A}_{s+1} \right) \right)$$

$$= \sum_{n_k=n_0}^{r} \sum_{n_1=n_k}^{r} \mathbb{P} \left( \hat{Y}_{1,n_1} \leq \xi \right) \sum_{s=n_0}^{r} \left( \sum_{\mathcal{S} \in [n_1, n_k]} \mathbb{P} \left( S_k^s(n_1, n_k) = \mathcal{S} \right) \right) \mathbb{E} \left( \mathbb{1} \left( N_k(s) = n_k, k \in \mathcal{A}_{s+1} \right) \right)$$

$$= \sum_{n_k=n_0}^{r} \sum_{n_1=n_k}^{r} \mathbb{P} \left( \hat{Y}_{1,n_1} \leq \xi \right) \underbrace{\mathbb{E} \sum_{s=n_0}^{r} \mathbb{1} \left( N_k(s) = n_k, k \in \mathcal{A}_{s+1} \right)}_{\leq 1}$$

$$\leq \sum_{n_k=n_0}^{r} \sum_{n_1=n_k}^{r} \mathbb{P} \left( \hat{Y}_{1,n_1} \leq \xi \right)$$

$$\leq r \sum_{n_1=n_k}^{r} \mathbb{P} \left( \hat{Y}_{1,n_1} \leq \xi \right) .$$

Here we have used the independence of the $S_k^s(m,n)$ from the reward streams and the fact that for every subset $\mathcal{S}$ of size $n_1$, $\hat{Y}_{k,n_1}$ and $\hat{Y}_{k,\mathcal{S}}$ have the same distribution. We can conclude as follows, proving the lemma:

$$\sum_{s=1}^{r} \mathbb{P}(\mathcal{Q}_k^r) \leq \sum_{s=1}^{r} \mathbb{P}(\mathcal{Q}_k^{r,1}) + \sum_{s=1}^{r} \mathbb{P}(\mathcal{Q}_k^{r,2})$$

$$\leq \sum_{n_k=n_0}^{r} \mathbb{P} \left( \hat{Y}_{k,n_k} \geq \xi \right) + r \sum_{n_1=n_0}^{r} \mathbb{P} \left( \hat{Y}_{1,n_1} \leq \xi \right) .$$

## D   Regret Decomposition: Proof of Lemma 4.1

We recall that we assume that arm 1 is the only optimal arm: $\mu_1 = \max_{k \in [K]} \mu_k$. The proof in this section follows the path of the proof in [2] for SSMC, but hinges on the new concentration result of Lemma 4.2. Moreover, some parts need to be adapted to handle the properties of an independent sampler instead of the duelling rule used in SSMC. As in [2], we introduce the following events:

- $\mathcal{G}_k^T = \cup_{r=1}^{T-1} \{\ell(r) = 1\} \cap \{k \in \mathcal{A}_{r+1}\} \cap \{N_k(r) \geq (1+\varepsilon)\xi_k \log T\}$

- $\mathcal{H}_k^T = \cup_{r=1}^{T-1} \{\ell(r) = 1\} \cap \{k \in \mathcal{A}_{r+1}\} \cap \{N_k(r) \geq J_k \log T\}$

- $\mathcal{Z}^r = \{\ell(r) \neq 1\}$, the leader used at round $r+1$ is sub-optimal.

These events directly provide an upper bound of the number of pulls of a sub-optimal arm $k$:

$$\mathbb{E}[N_k(T)] = \mathbb{E}[N_k(T) \mathbb{1}_{\mathcal{H}_k^T}] + \mathbb{E}[N_k(T) \mathbb{1}_{\mathcal{G}_k^T} \mathbb{1}_{\overline{\mathcal{H}_k^T}}] + \mathbb{E}[N_k(T) \mathbb{1}_{\overline{\mathcal{G}_k^T}}]$$

$$\leq T \mathbb{P}(\mathcal{H}_k^T) + (1 + J_k \log T) \mathbb{P}(\mathcal{G}_k^T) + 1 + (1+\varepsilon)\xi_k \log T + 2 \sum_{r=1}^{T-1} \mathbb{P}(\mathcal{Z}^r) \quad (3)$$

Indeed, due to the definition of each event we have:

$$N_k(T)\mathbb{1}_{\bar{\mathcal{G}}_k^T} \leq 1 + \sum_{r=1}^{T-1} \mathbb{1}_{(k \in \mathcal{A}_{r+1})} \mathbb{1}_{(\ell(r)\neq 1) \cup (N_k(r) < (1+\varepsilon)\xi_k \log(T))}$$

$$\leq 1 + \sum_{r=1}^{T-1} \mathbb{1}_{(k \in \mathcal{A}_{r+1})} \mathbb{1}_{(N_k(r) < (1+\varepsilon)\xi_k \log(T))} + \sum_{r=1}^{T-1} \mathbb{1}_{(\ell(r)\neq 1)}$$

$$\leq 1 + (1+\varepsilon)\xi_k \log T + \sum_{r=1}^{T-1} \mathbb{1}_{\mathcal{Z}^r}$$

and similarly

$$N_k(T)\mathbb{1}_{\mathcal{G}_k^T} \mathbb{1}_{\bar{\mathcal{H}}_k^T} \leq \left(1 + J_k \log T + \sum_{r=1}^{T-1} \mathbb{1}_{\mathcal{Z}^r}\right)\mathbb{1}_{\mathcal{G}_k^T}$$

$$\leq (1 + J_k \log T)\mathbb{1}_{\mathcal{G}_k^T} + \sum_{r=1}^{T-1} \mathbb{1}_{\mathcal{Z}^r}$$

Choosing $\xi_k = 1/I_1(\mu_k)$ the bound in (3) exhibits the term in $\frac{1+\varepsilon}{I_1(\mu_k)} \log T$ in Lemma 4.1. To obtain the result, it remains to upper bound

$$T\mathbb{P}(\mathcal{H}_k^T) + (1 + J_k \log T)\mathbb{P}(\mathcal{G}_k^T) + 2\sum_{r=1}^{T-1} \mathbb{P}(\mathcal{Z}^r)$$

for an appropriate choice of $J_k$. To do so, we shall first use the concentration inequality Lemma 4.2 to upper bound the terms involving $\mathcal{G}_k^T$ and $\mathcal{H}_k^T$ by problem-dependent constants (Appendix D.1), and then we carefully handle the terms in $\mathcal{Z}^r$ (Appendix D.2).

## D.1 Upper Bounds on $\mathbb{P}(\mathcal{G}_k^T)$ and $\mathbb{P}(\mathcal{H}_k^T)$

We first fix some real numbers $J_k$ and $\omega, \omega_k$ in $(\mu_k, \mu_1)$ to be specified later. We also fix $\xi_k = 1/I_1(\mu_k)$. Starting with $\mathcal{G}_k^T$, we apply the second statement in Lemma 4.2 for arm $k$ and arm 1 with $n_0 = (1+\varepsilon)\xi_k \log r$ and $\xi = \omega_k$:

$$\mathbb{P}(\mathcal{G}_k^T) \leq \sum_{r=1}^{T-1} \mathbb{P}\left(\ell(r) = 1, k \in \mathcal{A}_{r+1}, N_k(r) \geq (1+\varepsilon)\xi_k \log T\right)$$

$$\leq \sum_{r=1}^{T-1} \mathbb{P}\left(N_1(r) \geq N_k(r), \hat{Y}_{k,N_k(r)} > \hat{Y}_{1,S_1^r(N_1(r),N_k(r))}, N_k(r) \geq (1+\varepsilon)\xi_k \log T\right)$$

$$\leq \frac{T}{1 - e^{-I_1(\omega_k)}} e^{-(1+\varepsilon)\xi_k I_1(\omega_k) \log T} + \frac{1}{1 - e^{-I_k(\omega_k)}} e^{-(1+\varepsilon)\xi_k I_k(\omega_k) \log T}.$$

Similarly, we obtain

$$\mathbb{P}(\mathcal{H}_k^T) \leq \frac{T}{1 - e^{-I_1(\omega)}} e^{-J_k I_1(\omega) \log T} + \frac{1}{1 - e^{-I_k(\omega)}} e^{-J_k I_k(\omega) \log T}.$$

Our objective is to bound $T\mathbb{P}(\mathcal{H}_k^T)$ and $\log(T)\mathbb{P}(\mathcal{G}_k^T)$ by constants. This is achieved for instance if $T\mathbb{P}(\mathcal{H}_k^T) \xrightarrow[T\to+\infty]{} 0$ and $\log(T)\mathbb{P}(\mathcal{G}_k^T) \xrightarrow[T\to+\infty]{} 0$. The following conditions are sufficient to ensure these properties:

- $(1+\varepsilon)\xi_k I_1(\omega_k) > 1$
- $(1+\varepsilon)\xi_k I_k(\omega_k) > 0$
- $J_k I_1(\omega) > 2$
- $J_k I_k(\omega) > 1$

These conditions are met with the following values:

- $\omega = \frac{1}{2}(\mu_1 + \max_{k \neq 1} \mu_k)$

- $J_k > \max(\frac{1}{I_k(\omega)}, \frac{2}{I_1(\omega)})$

- $\mu_k < \omega_k < \mu_1$ chosen such that $(1+\varepsilon)I_1(\omega_k) > I_k(\omega_k)$. We are sure that such value exists if we choose $\omega_k$ close enough to $\mu_k$, thanks to the continuity of the rate functions and the fact that $I_k(\mu_k) = 0$ and $I_1(\mu_k) \neq 0$ (assumed in Assumption 1. of Theorem 3.1).

Choosing these values, both the terms in $\mathcal{G}_k^T$ and $\mathcal{H}_k^T$ in (3) are part of the constant $C_k(\boldsymbol{\nu}, \varepsilon)$ in Lemma 4.1. We can now focus on upper bounding $\sum_{r=1}^{T-1} \mathbb{P}(\mathcal{Z}^r)$, which is more challenging.

## D.2  Upper Bound on $\sum_{r=1}^{T-1} \mathbb{P}(\mathcal{Z}^r)$

The first steps of this part of the proof are again similar to [2]. The definition of the leader as the arm with the largest history gives the following property, that will be very useful for the analysis:

$$\ell(r) = k \Rightarrow N_k(r) \geq \left\lfloor \frac{r}{K} \right\rfloor - 1$$

So if an arm $k$ is the leader at a given round it has been drawn a linear amount of time at this round. Intuitively, this will provide very interesting concentration guarantees for the leader after a reasonable amount of rounds, that we are going to use in this section. For every $r \geq 8$, we define $a_r = \lfloor \frac{r}{4} \rfloor$ and use the decomposition

$$\mathbb{P}\left(\mathcal{Z}^r\right) = \mathbb{P}\left(\mathcal{Z}^r \cap \mathcal{D}^r\right) + \mathbb{P}\left(\mathcal{Z}^r \cap \bar{\mathcal{D}}^r\right), \tag{4}$$

where $\mathcal{D}^r$ is the event that the optimal has been leader at least once in $[a_r, r]$:

$$\mathcal{D}^r = \{\exists u \in [a_r, r] \text{ such that } \ell(u) = 1\}.$$

We now explain how to upper bound the sum of the two terms in the left hand side of (4).

### D.2.1   Controlling $\mathbb{P}(\mathcal{Z}^r \cap \mathcal{D}^r)$: arm 1 has been leader between $\lfloor r/4 \rfloor$ and $r$

We introduce a new event

$$\mathcal{B}^u = \{\ell(u) = 1, k \in \mathcal{A}_{u+1}, N_k(u) = N_1(u) - 1 \text{ for some arm } k\}$$

If $\mathcal{D}^r$ happens, then the event $\mathcal{Z}^r$ can be true only if the leadership has been taken over by a sub-optimal arm at some round between $a_r$ and $r$, that is

$$\mathcal{Z}^r \cap \mathcal{D}^r \subset \cup_{u=a_r}^r \{\bar{\mathcal{Z}}_u, \mathcal{Z}_{u+1}\} \subset \cup_{u=a_r}^r \mathcal{B}^u$$

We now upper bound $\sum_{r=8}^{T-1} \sum_{u=a_r}^r \mathbb{P}(\mathcal{B}^u)$. We use the notation $b_r = \lfloor a_r/K \rfloor$, where we recall $a_r = \lfloor r/4 \rfloor$. Then we write $\mathcal{B}^u = \cup_{k=2}^K \mathcal{B}_k^u := \{\ell(u) = 1, k \in \mathcal{A}_{u+1}, N_k(u) = N_1(u) - 1\}\}$, which fixes a specific suboptimal arm. For any $w_k$ in $(\mu_k, \mu_1)$, one can write

$$\sum_{r=8}^{T-1} \sum_{u=a_r}^r \mathbb{P}(\mathcal{B}_k^u) = \mathbb{E} \sum_{r=8}^{T-1} \sum_{u=a_r}^r \mathbb{1}(\ell(u) = 1)\mathbb{1}(k \in \mathcal{A}_{u+1})\mathbb{1}(N_1(u) = N_k(u) + 1)$$

$$\leq \mathbb{E} \sum_{r=8}^{T-1} \sum_{u=a_r}^r \mathbb{1}(N_1(u) \geq b_r)\mathbb{1}(\bar{Y}_{k,N_k(u)} \geq \bar{Y}_{1,S_k^u(N_1(u),N_k(u))})\mathbb{1}(N_1(u) = N_k(u) + 1)\mathbb{1}(k \in \mathcal{A}_{u+1})$$

$$\leq \mathbb{E} \sum_{r=8}^{T-1} \sum_{u=a_r}^r \mathbb{1}(N_1(u) \geq b_r)\mathbb{1}(\bar{Y}_{k,N_k(u)} < w_k)\mathbb{1}(N_1(u) = N_k(u) + 1)\mathbb{1}(k \in \mathcal{A}_{u+1}) \tag{5}$$

$$+ \mathbb{E} \sum_{r=8}^{T-1} \sum_{u=a_r}^r \mathbb{1}(N_1(u) \geq b_r)\mathbb{1}(\bar{Y}_{1,S_k^u(N_1(u),N_k(u))} > w_k)\mathbb{1}(N_1(u) = N_k(u) + 1)\mathbb{1}(k \in \mathcal{A}_{u+1}) \tag{6}$$

We now separately upper bound each of these two terms. First,

$$(5) \leq \mathbb{E} \sum_{r=8}^{T-1} \sum_{u=a_r}^{r} \sum_{n_k=b_r-1}^{r} \mathbb{1}(N_k(u) = n_k)\mathbb{1}(k \in \mathcal{A}_{u+1})\mathbb{1}(\bar{Y}_{k,n_k} < w_k)$$

$$\leq \mathbb{E} \sum_{r=8}^{T-1} \sum_{n_k=b_r-1}^{r} \mathbb{1}(\bar{Y}_{k,n_k} < w_k) \underbrace{\sum_{u=a_r}^{r} \mathbb{1}(N_k(u) = n_k)\mathbb{1}(k \in \mathcal{A}_{u+1})}_{\leq 1}$$

$$\leq \sum_{r=8}^{T-1} \sum_{n_k=b_r-1}^{r} \mathbb{P}(\bar{Y}_{k,n_k} < w_k)$$

$$\leq \sum_{r=8}^{T-1} \sum_{n_k=b_r-1}^{r} \exp\left(-n_k I_k(w_k)\right)$$

$$\leq \frac{e^{(2+1/K)I_k(\omega_k)}}{(1 - e^{-I_k(\omega_k)})(1 - e^{-I_k(\omega_k)/4K})}$$

Then, letting $[m, n]$ denote the set of subset of $[n]$ of size $m$,

$$(6) \leq \mathbb{E} \sum_{r=8}^{T-1} \sum_{u=a_r}^{r} \sum_{n_k=b_r-1}^{r} \mathbb{1}(\bar{Y}_{1,S_k^u(n_k+1,n_k)} > w_k)\mathbb{1}(N_k(u) = n_k)\mathbb{1}(k \in \mathcal{A}_{u+1})$$

$$\leq \mathbb{E} \sum_{r=8}^{T-1} \sum_{u=a_r}^{r} \sum_{n_k=b_r-1}^{r} \sum_{\mathcal{S} \in [n_k, n_k+1]} \mathbb{1}(\bar{Y}_{1,\mathcal{S}} > w_k)\mathbb{1}(S_k^u(n_k+1, n_k) = \mathcal{S})\mathbb{1}(N_k(u) = n_k)\mathbb{1}(k \in \mathcal{A}_{u+1})$$

$$\leq \mathbb{E} \sum_{r=8}^{T-1} \sum_{u=a_r}^{r} \sum_{n_k=b_r-1}^{r} \sum_{\mathcal{S} \in [n_k, n_k+1]} \mathbb{1}(\bar{Y}_{1,\mathcal{S}} > w_k)\mathbb{1}(N_k(u) = n_k)\mathbb{1}(k \in \mathcal{A}_{u+1})$$

$$\leq \mathbb{E} \sum_{r=8}^{T-1} \sum_{n_k=b_r-1}^{r} \sum_{\mathcal{S} \in [n_k, n_k+1]} \mathbb{1}(\bar{Y}_{1,\mathcal{S}} > w_k) \underbrace{\sum_{u=a_r}^{r} \mathbb{1}(N_k(u) = n_k)\mathbb{1}(k \in \mathcal{A}_{u+1})}_{\leq 1}$$

$$\leq \sum_{r=8}^{T-1} \sum_{n_k=b_r-1}^{r} \sum_{\mathcal{S} \in [n_k, n_k+1]} \mathbb{P}(\bar{Y}_{1,\mathcal{S}} > w_k)$$

$$\leq \sum_{r=8}^{T-1} \sum_{n_k=b_r-1}^{r} (n_k + 1)\mathbb{P}(\bar{Y}_{1,n_k} > w_k)$$

$$\leq \sum_{r=8}^{T-1} (r + 1) \sum_{n_k=b_r-1}^{r} \exp\left(-n_k I_1(w_k)\right)$$

$$\leq \frac{e^{(2+1/K)I_1(\omega_k)}}{(1 - e^{-I_1(\omega_k)})(1 - e^{-I_1(\omega_k)/4K})^2}$$

Here we have used that there are $n_k + 1$ subsets in $[n_k, n_k + 1]$ and that $\mathbb{P}(\bar{Y}_{1,\mathcal{S}} > w_k) = \mathbb{P}(\bar{Y}_{1,n_k} > w_k)$ for all such subsets. Choosing $\omega_k$ such that $I_1(\omega_k) = I_k(\omega_k)$ (which is possible given the continuity assumptions on the two rate functions), we obtain

$$\sum_{r=8}^{T-1} \mathbb{P}\left(\mathcal{Z}^r \cap \mathcal{D}^r\right) \leq \sum_{r=8}^{T-1} \sum_{u=a_r}^{r} \mathbb{P}(\mathcal{B}^u) \leq \sum_{k=2}^{K} \frac{2e^{(2+1/K)I_1(\omega_k)}}{(1 - e^{-I_1(\omega_k)})(1 - e^{-I_1(\omega_k)/4K})^2} . \tag{7}$$

### D.2.2 Controlling $\mathbb{P}(\mathcal{Z}^r \cap \mathcal{D}^r)$: arm 1 has not been leader between $\lfloor r/4 \rfloor$ and $r$

The idea in this part is to leverage the fact that if the optimal arm is not leader between $\lfloor s/4 \rfloor$ and $s$, then it has necessarily lost a lot of duels against the current leader at each round. We then use the

fact that when the leader has been drawn "enough", concentration prevents this situation with large probability. We introduce

$$\mathcal{L}^r = \sum_{u=s_0}^{r} \mathbb{1}_{\mathcal{C}^u}$$

for the event $\mathcal{C}^u = \{\exists a \neq 1, N_a(u) \geq N_1(u), \hat{Y}_{a,S_1^u(N_a(u),N_1(u))} \geq \hat{Y}_{1,N_1(u)}\}$. One can prove the following inequality:

$$\mathbb{P}(\mathcal{Z}^r \cap \bar{\mathcal{D}}^r) \leq \mathbb{P}(\mathcal{L}^r \geq r/4) .$$

*Proof.* Under $\bar{\mathcal{D}}^r$ arm 1 is a challenger for every round $u \in [a_r, r]$. Then, each time $\mathcal{C}^u$ is not true arm 1 wins its duel against the current leader and is pulled. Hence, if $\{\mathcal{L}_r < r/4\}$ then we necessarily have $\{N_1(r) > r/2\}$ and arm 1 is leader in round $r$. Hence, $\{\mathcal{Z}^r \cap \bar{\mathcal{D}}^r\} \cap \{\mathcal{L}_r < r/4\} = \emptyset$, which justifies the inequality. $\qquad\square$

Now, as in [2] we use the Markov inequality to get:

$$\mathbb{P}(\mathcal{L}^r \geq r/4) \leq \frac{\mathbb{E}(\mathcal{L}^r)}{r/4} = \frac{4}{r} \sum_{u=\lfloor r/4 \rfloor}^{r} \mathbb{P}(\mathcal{C}^u) .$$

By further decomposing the probability of $\mathbb{P}(\mathcal{C}^u)$ in two parts depending on the value of the number of selections of arm 1, we obtain the upper bound

$$\mathbb{P}(\mathcal{Z}^r \cap \overline{\mathcal{D}}^r) \leq \frac{4}{r} \sum_{u=\lfloor r/4 \rfloor}^{r} \mathbb{P}\left(N_1(u) \leq (\log u)^2\right) + \underbrace{\frac{4}{r} \sum_{u=\lfloor r/4 \rfloor}^{r} \mathbb{P}\left(\mathcal{C}^u, N_1(u) \geq (\log u)^2\right)}_{B_r} .$$

We now upper bound the quantity $B_r$ defined above by using Lemma 4.2. For each $a$, for any $\omega_a$ such that $\omega_a \in (\mu_a, \mu_1)$, one can write

$$B_r \leq \sum_{u=\lfloor r/4 \rfloor}^{r} \mathbb{P}(\mathcal{C}^u, N_1(u) \geq (\log\lfloor r/4 \rfloor)^2)$$

$$\leq \sum_{a=2}^{K} \sum_{u=\lfloor r/4 \rfloor}^{r} \mathbb{P}(Y_{a,S_1^u(N_a(u),N_1(u))} > \hat{Y}_{1,N_1(u)}, N_1(u) \geq \log(\lfloor r/4 \rfloor)^2, N_a(u) > N_1(u))$$

$$\leq \sum_{a=2}^{K} \left( \frac{1}{1-e^{-I_1(\omega_a)}} e^{-(\log\lfloor r/4 \rfloor)^2 I_1(\omega_a)} + \frac{r}{1-e^{-I_k(\omega_a)}} e^{-(\log\lfloor r/4 \rfloor)^2 I_a(\omega_a)} \right) .$$

Choosing each $\omega_a$ such that $I_1(\omega_a) = I_a(\omega_a)$, we obtain:

$$\frac{4}{r} \sum_{r=8}^{T} B_r \leq \sum_{r=8}^{T} \sum_{a=2}^{K} \frac{4(r+1)}{r(1-e^{-I_1(\omega_a)})} e^{-(\log\lfloor r/4 \rfloor)^2 I_a(\omega_a)}$$

$$\leq \sum_{a=2}^{K} \sum_{r=8}^{T} \frac{6}{1-e^{-I_1(\omega_a)}} e^{-(\log\lfloor r/4 \rfloor)^2 I_a(\omega_a)},$$

and for each $a$ the series in $r$ is convergent as for any constant $C$, $C \log(r) \leq (\log\lfloor r/4 \rfloor)^2$ for $r$ large enough. Hence, there exists some constant $D(\boldsymbol{\nu})$ where $\boldsymbol{\nu} = (\nu_1, \ldots, \nu_K)$ such that $\frac{4}{r} \sum_{r=8}^{T} B_r \leq D(\boldsymbol{\nu})$. It follows that

$$\sum_{r=8}^{T} \mathbb{P}(\mathcal{Z}^r \cap \overline{\mathcal{D}}^r) \leq \sum_{r=8}^{T} \frac{4}{r} \sum_{u=\lfloor r/4 \rfloor}^{r} \mathbb{P}\left(N_1(u) \leq (\log u)^2\right) + D(\boldsymbol{\nu}) .$$

We now transform the double sum in the right-hand side into a simple sum by counting the number of times each term appears in the double sum:

$$\sum_{r=8}^{T} \frac{4}{r} \sum_{u=\lfloor r/4 \rfloor}^{r} \mathbb{P}\left(N_1(u) \leq (\log u)^2\right) = \sum_{r=8}^{T} \left( \sum_{t=1}^{r} \frac{4}{t} \mathbb{1}(t \in [r, 4r]) \right) \mathbb{P}(N_1(r) \leq (\log r)^2) .$$

Noting that $\sum_{t=1}^{r} \frac{4}{t} \mathbb{1}(t \in [s, 4s]) \leq (4s - s + 1) \times \frac{4}{s} \leq 16$, we finally obtain:

$$\sum_{r=8}^{T} \mathbb{P}(\mathcal{Z}^r \cap \overline{\mathcal{D}}^r) \leq 16 \sum_{r=1}^{T} \mathbb{P}\left(N_1(r) \leq (\log(r))^2\right) + D(\boldsymbol{\nu}). \tag{8}$$

Combining (7) and (8) yields

$$\sum_{r=1}^{T} \mathbb{P}(\mathcal{Z}^r) \leq 16 \sum_{r=1}^{T} \mathbb{P}\left(N_1(r) \leq (\log(r))^2\right) + D'_k(\boldsymbol{\nu})$$

for some constant $D'_k(\boldsymbol{\nu})$ that depends on $k$ and $\boldsymbol{\nu}$, which contributes to the final constant $C_k(\boldsymbol{\nu}, \varepsilon)$ in Lemma 4.1. Plugging this inequality in Equation (3) concludes the proof of Lemma 4.1.

# E  Probability that the Optimal Arm is not Drawn Enough: Proof of Lemma 4.3

We start with a decomposition that follows the steps of [1] for BESA with 2 arms that we generalize for $K$ arms.

We first denote by $r_j$ the round of the $j^{th}$ play of arm 1 with $r_0 = 0$ and let $\tau_j = r_{j+1} - r_j$. We notice that $\tau_0 \leq K$ as all arms are initialized once. Then:

$$\mathbb{P}\left(N_1(r) \leq (\log r)^2\right) \leq \mathbb{P}\left(\exists j \in \{1, ..., \log r^2\} : \tau_j \geq r/(\log r)^2 - 1\right)$$

$$\leq \sum_{j=1}^{(\log r)^2} \mathbb{P}\left(\tau_j \geq r/(\log r)^2 - 1\right)$$

*Proof.* If we assume that $\forall j \; \tau_j \leq r/(\log r)^2 - 1$ then $t_{\log r^2} = \sum_{j=0}^{\log r^2} \tau_j < r$, which yields $N_\ell(r) > \log r^2 + 1$. $\qquad\square$

We now fix $j \leq (\log r)^2$ and upper bound the probability of the event

$$\mathcal{E}_j := \{\tau_j \geq r/\log r^2 - 1\}.$$

On this event arm 1 lost at least $r/\log r^2$ consecutive duels between $r_j + 1$ and $r_{j+1}$ (either as a challenger of as the leader) which yields

$$\mathbb{P}(\mathcal{E}_j) \leq \mathbb{P}\left(\forall s \in \{r_j + 1, ..., r_j + \lfloor r/\log r^2 - 1\rfloor\} : \{\hat{Y}_{1,j} \leq \hat{Y}_{\ell(s), S_1^s(N_{\ell(s)}(s), j)}, N_1(s) = j, N_{\ell(s)}(s) \geq j\}\right.$$
$$\left.\cup \{\ell(s) = 1, N_1(s) = j\}\right)$$

The important change compared to the proof of [1] is that with $K > 2$, 1) we don't know the identity of the leader and 2) the leader is not necessarily pulled if it wins its duel against 1.

Now we notice that when $r$ is large, the time range considered in $\mathcal{E}_j$ is large. By looking at the second half of this time range only, we can ensure that the leader has been drawn a large number of times. More precisely, introducing the two intervals

$$\mathcal{M}_{j,r}^1 = \left[r_j + 1, r_j + \left\lfloor \frac{r/\log r^2 - 1}{2}\right\rfloor\right]$$

$$\mathcal{M}_{j,r}^2 = \left[t_j + \left\lceil \frac{r/\log r^2 - 1}{2}\right\rceil, t_j + \lfloor r/\log r^2\rfloor - 1\right]$$

it holds that

$$\mathbb{P}(\mathcal{E}_j) \leq \mathbb{P}(\forall s \in \mathcal{M}_{j,r}^2 : \{\hat{Y}_{1,j} \leq \hat{Y}_{\ell(s), S_1^s(N_{\ell(s)}(s), j)}, N_1(s) = j, N_{\ell(s)}(s) \geq j\} \cup \{\ell(s) = 1, N_1(s) = j\}).$$

But we know that on $\mathcal{M}_{j,r}^2$ the leader must has been selected at least $\frac{1}{K}\left(j + \left\lceil \frac{r/\log r^2 - 1}{2}\right\rceil\right)$ times.

Let $r_K$ be the first integer such that $\log^2(r) < \frac{1}{K-1}\left\lceil \frac{r/\log r^2 - 1}{2}\right\rceil$, for every $r \geq r_K$, as $j \leq \log^2(r)$,

the leader has been selected strictly more than $j$ times, which prevents arm 1 from being the leader for any round in $\mathcal{M}_{j,r}^2$. Hence, for $r \geq r_K$, for all $j \leq \log^2(r)$,

$$\mathbb{P}(\mathcal{E}_j) \leq \mathbb{P}\left(\forall s \in \mathcal{M}_{j,r}^2 : \{\hat{Y}_{1,j} \leq \hat{Y}_{\ell(s),S_1^s(N_{\ell(s)}(s),j)}, N_1(s) = j, N_{\ell(s)}(s) \geq j\}\right).$$

To remove the problem of the identity of the leader we would like to find a way to fix our attention on one arm. To this extent, we notice that during an interval of length $|\mathcal{M}_{j,r}^2|$, if there are only $K-1$ candidates for the leader then one of them must have been leader at least $m_r := |\mathcal{M}_{j,r}^2|/(K-1) - 1$ times during this range. We also know that at any round in $\mathcal{M}_{j,r}^2$, the leader satisfies $N_{\ell(s)}(s) \geq (t_j + \lfloor \frac{r/\log r^2 - 1}{2} \rfloor)/K - 1 \geq (\lfloor \frac{r/\log r^2 - 1}{2} \rfloor)/K - 1 = \frac{|\mathcal{M}_{j,r}^1|}{K} - 1 := c_r$. Observe that $m_r > c_r$. Finally, we introduce the notation

$$I_{j,r}^k = \{s \in \mathcal{M}_{j,r}^2 : \ell(s) = k\}$$

for the set of rounds in $\mathcal{M}_{j,r}^2$ in which a particular arm $k$ is leader. From the above discussion, we know that there exists an arm $k$ such that $|I_{j,r}^k| \geq m_r$.

To ease the notation, we introduce the event

$$\mathcal{W}_{s,j}^k = \left\{\left\{\hat{Y}_{1,j} < \hat{Y}_{k,S_1^s(N_k(s),j)}\right\}, N_k(s) \geq c_r, N_1(s) = j\right\}$$

and write

$$\mathbb{P}(\mathcal{E}_j) \leq \mathbb{P}\left(\bigcap_{s \in \mathcal{M}_{j,r}^2} \bigcup_{k=2}^K \{\ell(s) = k, 1 \notin \mathcal{A}_s)\}\right)$$

$$\leq \mathbb{P}\left(\bigcap_{k=2}^K \bigcap_{s \in I_{j,r}^k} \mathcal{W}_{s,j}^k\right)$$

$$\leq \mathbb{P}\left(\bigcup_{k=2}^K \left\{|I_{j,r}^k| > m_r, \bigcap_{s \in I_{j,r}^k} \mathcal{W}_{s,j}^k\right\}\right)$$

$$\leq \sum_{k=2}^K \mathbb{P}\left(|I_{j,r}^k| > m_r, \bigcap_{s \in I_{j,r}^k} \mathcal{W}_{s,j}^k\right).$$

Finally, we define for any integer $M$ the event that we can find $M$ pairwise non-overlapping sub-samples in the set of the sub-samples of arm $k$ drawn in rounds $s \in I_{j,r}^k$:

$$\mathcal{F}_{j,M}^{k,r} = \left\{\exists i_1, ..., i_M \in I_{j,r}^k : \forall m < m' \in [M], S_1^{i_m}(N_k(i_m), j) \cap S_1^{i_{m'}}(N_k(i_{m'}), j) = \emptyset\right\}$$

Introducing $H_{j,r}^k = \min_{s \in I_{j,r}^k} N_k(s)$, the minimal size of the history of arm $k$ during rounds in $I_{j,r}^k$ (which is known to be larger than $c_r$ as $k$ is leader in these rounds), one has

$$\mathbb{P}(\mathcal{E}_j) \leq \sum_{k=2}^K \mathbb{P}\left(|I_{j,r}^k| > m_r, \cap_{s \in I_{j,r}^k} \mathcal{W}_{s,j} \cap \{\mathcal{F}_{j,M}^{k,r} \cup \bar{\mathcal{F}}_{j,M}^{k,r}\}\right)$$

$$\leq \sum_{k=2}^K \mathbb{P}\left(|I_{j,r}^k| \geq m_r, H_{j,r}^k \geq c_r, \bar{\mathcal{F}}_{j,M}^{k,r}\right) + \sum_{k=2}^K \mathbb{P}\left(|I_{j,r}^k| > m_r, \cap_{s \in I_{j,r}^k} \mathcal{W}_{s,j} \cap \mathcal{F}_{j,M}^{k,r}\right) \quad (9)$$

**Upper bound on the first term in** (9)   The probability $\mathbb{P}\left(|I_{j,r}^k| \geq m_r, H_{j,r}^k \geq c_r, \bar{\mathcal{F}}_{j,M}^{k,r}\right)$ can be upper bounded by

$$\mathbb{P}\left(\#\left\{\text{pairwise non-overlapping subsets in } (S_1^s(N_k(s), j))_{s \in I_{j,r}^k}\right\} < M \,\middle|\, \{|I_{j,r}^k| > m_r, H_{j,r}^k \geq c_r\}\right).$$

This probability can be related to some intrinsic properties of the sampler $\mathrm{SP}(H, j)$. To formalize this, we introduce the following definition.

**Definition.** *For every integers $N, H, j$ such that $H > j$, $X_{N,H,j}$ is a random variable which counts the maximum number of non-overlapping subsets among $N$ i.i.d. samples from $SP(H,j)$.*

Letting $H_1, \ldots, H_{m_r}$ be integers that are all larger than $c_r$, and letting $S_1, \ldots, S_{m_r}$ be independent subsets such that $S_i \sim SP(H_i, j)$, the above probability is upper bounded by

$$\mathbb{P}\left(\# \{\text{pairwise non-overlapping subsets in } (S_i)_{i=1}^{m_r}\} < M\right)$$

which is itself upper bounded by $\mathbb{P}\left(X_{m_r, c_r, j} < M\right)$.

This last inequality is quite intuitive: if one draws subsets of size $j$ from histories that may be larger than $c_r$, there is more "room" for non-overlapping subsets than if we always draw them from the same history of size $c_r$. For Random Block sampling, where the drawn subset is fully determined by the random position of its first element, to formalize this intuition it is sufficient to prove that if $X_i, Y_i$ are two sequences of random variables such that $X_i$ is uniform in $[H_i - j]$ and $Y_i$ is uniform in $[H - j]$, where $H_i \geq H$, the random variable that counts the maximal number of elements in the sequence $(Y_i)$ whose pairwise distance are larger than $j$ is stochastically dominated by that the same random variable but for the sequence $(X_i)$. We performed numerical experiments that confirm that this last condition holds.

**Upper bound on the second term in** (9) On the event $\left(|I_{j,r}^k| > m_r, \cap_{s \in I_{j,r}^k} \mathcal{W}_{s,j} \cap \mathcal{F}_{j,M}^{k,r}\right)$, one can define $\tilde{i}_1, \ldots, \tilde{i}_M$ the first $M$ rounds in $I_{j,r}^k$ for which the subsets $\tilde{S}_m := S^{\tilde{i}_m}(N_k(\tilde{i}_m), j)$ are pairwise non-overlapping and we get

$$\mathbb{P}\left(|I_{j,r}^k| > m_r, \cap_{s \in I_{j,r}^k} \mathcal{W}_{s,j} \cap \mathcal{F}_{j,M}^{k,r}\right) \leq \mathbb{P}\left(\forall m \in [M], \hat{Y}_{1,j} \leq \hat{Y}_{k,\tilde{S}_m}\right) .$$

By definition the subsets $\tilde{S}_m$ are pairwise non-overlapping, hence the sub-samples $\hat{Y}_{k,\tilde{S}_m}$ are independent. We prove that this probability can be in fact upper bound by the *balance function* we defined in section 3.

Indeed, introducing $X \sim \nu_{1,j}$ and an independent i.i.d. sequence $Z_i \sim \nu_{k,j}$, one can write

$$
\begin{aligned}
\mathbb{P}\left(|I_{j,r}^k| > m_r, \cap_{s \in I_{j,r}^k} \mathcal{W}_{s,j} \cap \mathcal{F}_{j,M}^{k,r}\right) &\leq \mathbb{P}(X < \min_{i \in [M]} Z_i) \\
&= \mathbb{E}_{\substack{X \sim \nu_{1,j} \\ Z \sim \nu_{k,j}^{\otimes j}}} \left[\prod_{i=1}^{M} \mathbb{1}_{X \leq Z_i}\right] \\
&= \mathbb{E}_{X \sim \nu_{1,j}}\left[\mathbb{E}_{Z \sim \nu_{k,j}^{\otimes j}}\left[\prod_i \mathbb{1}_{X \leq Z_i} \,\middle|\, X\right]\right] \\
&= \mathbb{E}_{X \sim \nu_{1,j}}\left[(1 - F_{k,j}(X))^M\right] \\
&= \alpha_k(M, j).
\end{aligned}
$$

**Conclusion** Putting things together, we have proved that

$$\mathbb{P}(\mathcal{E}_j) \leq (K-1)\mathbb{P}\left(X_{m_r, c_r, j} < M\right) + \sum_{k=2}^{K} \alpha_k(M, j),$$

where $X_{N,H,j}$ and $\alpha_k(M, j)$ are introduced in Definition E and 3 respectively. If we replace $M$ by the sequence $\beta_{r,j}$ we have

$$
\begin{aligned}
\sum_{r=1}^{T} \mathbb{P}(N_1(r) \leq \log r^2) &\leq r_K + \sum_{r=r_K}^{T} \sum_{j=1}^{\log r^2}\left[(K-1)\mathbb{P}\left(X_{m_r, c_r, j} < \beta_{r,j}\right) + \sum_{k=2}^{K} \alpha_k(\beta_{r,j}, j)\right] \\
&\leq r_K + \sum_{r=r_K}^{T} \sum_{j=1}^{\log r^2}\left[(K-1)\mathbb{P}\left(X_{c_r, c_r, j} < \beta_{r,j}\right) + \sum_{k=2}^{K} \alpha_k(\beta_{r,j}, j)\right]
\end{aligned}
$$

as $c_r \leq m_r$, which proves Lemma 4.3.

This definition allows to analyze separately the properties of the sub-sampling algorithms and the properties of the distribution family for randomized samplers.

## F    Proof that RB-SDA Satisfies the Diversity Property

We recall that $X_{m,H,j}$ denotes the maximal number of pairwise non-overlapping subsets obtained in $m$ i.i.d. samples from $\text{RB}(H,j)$. In this section we aim at upper bounding the probability of

$$\mathbb{P}\left(X_{m,H,j} \leq \gamma r/(\log r)^2\right)$$

for some values of $m$, $H$, $j$, that will be fixed later. This probability depends on several parameters, with straightforward effects:

- The probability decreases with the length of the history size $H$.
- The probability increases with the size $j$ of each sub-sample.
- The probability decreases with the total number of sub-samples we draw $m$. Intuitively if $m$ is large enough every sample of size $j$ in the history will be drawn.

**First step with $j = 1$:**  in this case the distribution of the $m$ subsets of size 1 is actually the distribution of sampling with replacement in $H$. The question of the number of different items drawn with sampling without replacement has been studied in [25], from which we use the following result:

**Result 1**: for any $k \in [H]$, $\mathbb{P}(X_{m,H,1} = k) = \frac{H!}{(H-k)! \times H^m} \times S_{k,m}$, where $S_{k,m}$ is the Stirling number of the second kind for $k, m$.

We use this result with further assumptions that are specific to our problem and will ease the computation: $H = m = O(r/(\log r)^2)$. To ease the notation we continue to use $H$, and write $\gamma t/(\log t)^2 = \alpha H$ for some $\alpha \in (0,1)$.

We first look at $\mathbb{P}(X_{H,H,1} = \alpha H)$. According to [26] the following inequality holds

$$S_{k,H} \leq \frac{1}{2}\binom{H}{k}k^{H-k} \,,$$

This allows to upper bound the expression in result 1:

$$\mathbb{P}(X_{H,H,1} = k) \leq \frac{1}{2}\left(\frac{k}{H}\right)^{H-k}\binom{H}{k}$$

We now want to bound $\binom{H}{k}$. As $k$ is small compared with $H$, it is natural to use

$$\binom{H}{k} \leq \frac{H^k}{k!}$$

We then bound $1/k!$ by its Stirling approximation and add a multiplicative constant $c$ along the way:

$$\binom{H}{k} \leq c\frac{H^k}{\sqrt{2\pi k} \times k^k}e^k$$

Refactoring provides

$$\mathbb{P}(X_{H,H,1} = k) \leq \frac{c}{2}\left(\frac{k}{H}\right)^{H-2k}\frac{e^k}{\sqrt{2\pi k}}$$

Then we notice that if $k \leq H - 2k$ we get:

$$\mathbb{P}(X_{H,H,1} = k) \leq \frac{c}{2\sqrt{2\pi k}}\left(\frac{ke}{H}\right)^{H-2k}$$

Now we can replace $k$ by $\alpha H$ (assume it's an integer for the simplicity of notations), such that 1) $\alpha \leq \frac{1}{3} \Rightarrow H(1 - 3\alpha) > 0$ and $\alpha e < 1$.

If $k \le \alpha H$, $\alpha e \le 1$ and $(1 - 2\alpha) > 0$ then: $\left(\frac{ke}{H}\right)^{H-2k} \le (\alpha e)^{H-2k}$. We have:

$$
\begin{aligned}
\mathbb{P}(X_{H,H,1} \le \alpha H) &\le \frac{c}{2\sqrt{2\pi}} \sum_{k=0}^{\lfloor \alpha H \rfloor} (\alpha e)^{H-2k} \\
&\le \frac{c}{2\sqrt{2\pi}} \sum_{k=0}^{\lfloor \alpha H \rfloor} (\alpha e)^{H-2(\lfloor \alpha H \rfloor - k))} \\
&\le \frac{c}{2\sqrt{2\pi}} (\alpha e)^{H-2\lfloor \alpha H \rfloor} \frac{1}{1-(\alpha e)^2} \\
&\le \frac{c}{2\sqrt{2\pi}} \frac{1}{1-(\alpha e)^2} \exp\left(-(1-2\alpha)H\log(1/(\alpha e))\right)
\end{aligned}
\tag{10}
$$

**From $X_{H,H,1}$ to $X_{H,H,j}$**    This result is enough to get general properties for Random Block Sampling. Indeed, as the process of RBS consists in only drawing the first element of the block used in the duel we can see that the previous bound also applies to the number of unique starting points. With this property, the Random Block Sampler satisfies for all $x > 0$:

$$
\mathbb{P}\left(X_{m,H,j} \le \left\lfloor \frac{x}{j} \right\rfloor\right) \le \mathbb{P}\left(X_{m,H,1} \le x\right)
$$

*Proof.* Assume that the Random Block Sampler provides $x$ blocks with different starting points. Let's further assume that $x$ is an integer and try to identify the sequence of starting times $t_i = (t_1, ..., t_x)$ that minimizes the number of mutually non-overlapping samples: the value of $t_1$ is not important due to the symmetry of the problem. Then if we want to reduce the possibilities to get non-overlapping sample we want to choose a value for $t_2$ that 1) makes the blocks $[t_1, t_1 + j]$ and $[t_2, t_2 + k]$ non-overlapping and 2) makes things easier to continue this process for $t_3, ..., t_m$. It seems intuitive to choose either the block starting at $t_1 + 1$ or at $t_1 - 1$ as we cover the minimum amount of space with the constraint that $t_2 \ne t_1$. If we repeat this choice until $m$ blocks are chosen and reorder the blocks properly, we get a sequence of starting points $[t_1, t_1 + 1, ... t_1 + m]$ that are all different and minimize the total amount of space covered by the block. Even in this setup, we can find exactly $\left\lfloor \frac{m}{j} \right\rfloor$ mutually non-overlapping blocks as for instance all $[t_1 + kj, t_1 + (k+1)j - 1]$, $[t_1 + k'j, t_1 + (k'+1)j - 1]$ blocks are non-overlapping for $k \ne k'$ and $(k, k') \in [0, \left\lfloor \frac{m}{j} \right\rfloor - 1]$. $\square$

We can finally prove the following for Random Block sampling.

**Lemma F.1** (Diversity Property for Random Block Sampling). *If we choose a constant $\gamma \le 1/3 \times \frac{1}{2K}$ then Random Block Sampling satisfies the diversity property.*

*Proof.* For $\gamma \le \left\lfloor 1/3 \times \frac{1}{2K} \right\rfloor$, there exists $\alpha > 0$ such that:

$$
\begin{aligned}
\mathbb{P}(X_{c_r, c_r, j} \le \gamma/j(r/(\log r)^2)) &\le \mathbb{P}(X_{c_r, c_r, j} \le \alpha/jc_r) \\
&\le \mathbb{P}(X_{c_r, c_r, 1} \le \alpha c_r) \\
&= o(r^{-2})
\end{aligned}
$$

The last line comes from the expression obtained in Equation (10), and allows to conclude that $\sum_{r=1}^{T} \sum_{j=1}^{(\log r)^2} \mathbb{P}(X_{c_r, c_r, j} \le \alpha/j(r/(\log r))^2) = o(\log T)$ $\square$

# G   Analysis of the Balance Function for Some Distributions

For the simplicity of the notation we write the balance function $\alpha(M, j)$ for any distribution and any instance of these distributions. The family of distributions and the notation for their parameter will always mentioned at the beginning of the corresponding sub-section.

In the next parts we use the notation $G(x) = 1 - F(x)$ where $F$ denotes the CDF of some distribution. For some arm distribution $\nu_i$ the distribution of the sum of $j$ independent observations drawn from $\nu_i$ is denoted by $\nu_{i,j}$. With this notation, for two arms 1 and 2 we write:

$$
\alpha(M, j) = \mathbb{E}_{Z \sim \nu_{1,j}}(G_{2,j}(Z)^M)
$$

## G.1 The Bernoulli Distribution is Balanced

We prove the following lemma, which bears strong similarity with an upper bound given by [10] for a similar quantity in their analysis of Thompson Sampling.

**Lemma G.1** (Bound on $\alpha(M, j)$). *For two Binomial distributions $\nu_1 \sim \mathcal{B}(j, \mu_1)$ and $\nu_2 \sim \mathcal{B}(j, \mu_2)$ such that $\mu_1 > \mu_2$ and for any integer $M > 1$: $\exists \lambda > 1$ such as*

$$\mathbb{E}_{X \sim \nu_{1,j}} \left( (1 - F_{j,\mu_2}(X))^M \right) \leq C_{\lambda_0, \lambda} \frac{1}{M^\lambda} e^{-j d_{\lambda, \mu_1, \mu_2}} + \left( \frac{1}{2} \right)^M$$

*Where $C_{\lambda, \mu_1, \mu_2} > 0$, and $F_{j, \mu_2}$ is the CDF of a Binomial $\mathcal{B}(j, \mu_2)$.*

*Proof.* We use the same notation as before: $G_2(k) = 1 - F_2(k)$ and $f_1, f_2$ as the PMF of $\nu_1, \nu_2$.

We first use a common property of Binomial distributions, $\forall k > \lceil j\mu_2 \rceil$: $G(k) \leq \frac{1}{2}$. So we can directly write:

$$\mathbb{E}_{X \sim \nu_1} \left( (1 - F_{j,\mu_2}(X))^M \right) \leq \left( \frac{1}{2} \right)^M + \underbrace{\sum_{k=0}^{\lfloor j\mu_2 \rfloor} f_1(k) G_2(k)^M}_{(A)}$$

Using convexity we get: $G(k)^M \leq \exp\left( -M F_2(k) \right)$, hence

$$(A) \leq \sum_{k=0}^{\lfloor j\mu_2 \rfloor} f_1(k) \exp\left( -M F_2(k) \right)$$

Then we use that for $\lambda > 1$, $\forall x > 0$: $x^\lambda e^{-x} \leq \left( \frac{\lambda}{e} \right)^\lambda = C_\lambda$, so:

$$(A) \leq \frac{C_\lambda}{M^\lambda} \sum_{k=0}^{\lceil j\mu_2 \rceil} \frac{f_1(k)}{F_2(k)^\lambda} \leq \frac{C_\lambda}{M^\lambda} \sum_{k=0}^{\lceil j\mu_2 \rceil} \frac{f_1(k)}{f_2(k)^\lambda}$$

As in [10], we compute:

$$\begin{aligned}
\frac{f_1(k)}{f_2(k)^\lambda} &\leq \frac{\mu_1^k (1 - \mu_1)^{j-k}}{\mu_2^{\lambda k} (1 - \mu_2)^{\lambda(j_k)}} \\
&\leq \left( \frac{1 - \mu_1}{(1 - \mu_2)^\lambda} \right)^j \left( \frac{\mu_1 (1 - \mu_2)^\lambda}{\mu_2^\lambda (1 - \mu_1)} \right)^k \\
&= \left( \frac{1 - \mu_1}{(1 - \mu_2)^\lambda} \right)^j R_\lambda(\mu_1, \mu_2)^k
\end{aligned}$$

with $R_\lambda(\mu_1, \mu_2) = \frac{\mu_1 (1 - \mu_2)^\lambda}{\mu_2^\lambda (1 - \mu_1)}$. We then notice that we can choose $\lambda > 1$ such that $R_\lambda(\mu_1, \mu_2) > 1$. It is true for any $\lambda > 1$ if $\mu_2 \leq 0.5$, and for $1 < \lambda < \log \frac{\mu_1}{1 - \mu_1} / \log \frac{\mu_2}{1 - \mu_2}$ if $\mu_2 > 0.5$.

Plugging that expression into the sum gives:

$$
\begin{aligned}
(A) \leq & \frac{C_\lambda}{M^\lambda} \sum_{k=0}^{\lceil j\mu_2 \rceil} \frac{f_1(k)}{f_2(k)^\lambda} \\
\leq & \frac{C_\lambda}{M^\lambda} \left( \frac{1-\mu_1}{(1-\mu_2)^\lambda} \right)^j \sum_{k=0}^{\lceil j\mu_2 \rceil} R_\lambda(\mu_1,\mu_2)^k \\
= & \frac{C_\lambda}{M^\lambda} \left( \frac{1-\mu_1}{(1-\mu_2)^\lambda} \right)^j \frac{R_\lambda(\mu_1,\mu_2)^{\lfloor j\mu_2 \rfloor +1} - 1}{R_\lambda(\mu_1,\mu_2) - 1} \\
\leq & \frac{C_\lambda}{M^\lambda} \left( \frac{1-\mu_1}{(1-\mu_2)^\lambda} \right)^j \frac{R_\lambda(\mu_1,\mu_2)}{R_\lambda(\mu_1,\mu_2) - 1} R_\lambda(\mu_1,\mu_2)^{j\mu_2} \\
= & \frac{C_\lambda}{M^\lambda} \frac{R_\lambda(\mu_1,\mu_2)}{R_\lambda(\mu_1,\mu_2) - 1} \left( \frac{1-\mu_1}{(1-\mu_2)^\lambda} \right)^{j(1-\mu_2)} \left( \frac{\mu_1}{\mu_2^\lambda} \right)^{j\mu_2} \\
= & \frac{C_\lambda}{M^\lambda} \frac{R_\lambda(\mu_1,\mu_2)}{R_\lambda(\mu_1,\mu_2) - 1} e^{-jd_\lambda(\mu_2,\mu_1)} \\
= & \frac{C_{\lambda,\mu_1,\mu_2}}{M^\lambda} e^{-jd_\lambda(\mu_2,\mu_1)}
\end{aligned}
$$

where $d_\lambda(\mu_2,\mu_1) = \lambda \left( \mu_2 \log \mu_2 + (1-\mu_2) \log(1-\mu_2) \right) - (\mu_2 \log \mu_1 + (1-\mu_2) \log(1-\mu_1)) = \text{KL}(\mu_2,\mu_1) - (\lambda-1)\text{H}(\mu_2)$, $\text{KL}(\mu_2,\mu_1)$ denotes the KL-divergence between $\nu_2$ and $\nu_1$, and $\text{H}(\mu_2) = \mathbb{E}_{X \sim \nu_{2,j}}(\log f_2(X))$. We need to choose $\lambda$ as:

$$
\lambda < 1 + \frac{\text{KL}(\mu_2,\mu_1)}{\text{H}(\mu_2)} = \lambda_0(\mu_1,\mu_2)
$$

Note that those quantities correspond to the Bernoulli distributions, the $j$ is not involved here. In [10], the authors explain that this condition is more restrictive than the previous one so we can state that $\forall \lambda < \lambda_0(\mu_1,\mu_2)$:

$$
\mathbb{E}_{X \sim \nu_{1,j}} \left( (1 - F_{j,\mu_2}(X))^M \right) \leq \left( \frac{1}{2} \right)^M + \frac{C_{\lambda,\mu_1,\mu_2}}{M^\lambda} e^{-jd_\lambda(\mu_2,\mu_1)}
$$

$\square$

This is enough to prove that the Bernoulli distribution is balanced by replacing $M$ by $\lfloor \beta t/(\log t)^2 \rfloor$ in the expression in Lemma G.1 and summing on $t$ and $j$. The power term is in $o(t(\log t^2))$, while the other term is the term of a convergent geometric series in $j$ multiplied by a term in $o(1/t)$ in $t$, which is enough to get the result.

### G.2 The Poisson Distribution is Balanced

We can actually use the same sketch of proof as for Bernoulli distributions, using that for 2 Poisson random variables:

$$
\frac{p_{1,j}(k)}{p_{2,j}(k)^\lambda} = e^{-j(\theta_1 - \lambda\theta_2)} \left( \frac{k!}{n^k} \right)^\lambda \left( \frac{\theta_1}{\theta_2^\lambda} \right)^k \leq e^{-j(\theta_1 - \lambda\theta_2)} \left( \frac{\theta_1}{\theta_2^\lambda} \right)^k
$$

So:

$$
\begin{aligned}
\sum_{k=0}^{d_{0,j}} \frac{p_{1,j}(k)}{p_{2,j}(k)^\lambda} & \leq e^{-j(\theta_1 - \lambda\theta_2)} \sum_{k=0}^{d_{0,j}} \left( \frac{\theta_1}{\theta_2^\lambda} \right)^k \\
& \leq \frac{\theta_2^\lambda}{|\theta_1 - \theta_2^\lambda|} e^{-j(\theta_1 - \lambda\theta_2)} \times \max \left\{ 1, \left( \frac{\theta_1}{\theta_2^\lambda} \right)^{d_{0,j}} \right\},
\end{aligned}
$$

where $d_{0,j} = \theta_2 j - 1$. Now we remark that if we choose $\lambda \in (1, \theta_1/\theta_2)$ we have 2 possibilities: 1) we can choose $\lambda$ such that the second term equals one, hence we can bound the whole term by a constant without further conditions, or 2) $\forall \lambda \in (0, \theta_1, \theta_2)$: $\left(\frac{\theta_1}{\theta_2^\lambda}\right) > 1$. Let us focus on the second case, we study the term $e^{-j((\theta_1 - \lambda\theta_2) - \theta_2(\log\theta_1 - \lambda\log\theta_2))}$. As for Bernoulli distributions, we identify the KL-divergence between $\nu_2$ and $\nu_1$ and write:

$$(\theta_1 - \lambda\theta_2) - \theta_2(\log\theta_1 - \lambda\log\theta_2) = \mathrm{KL}(\nu_2, \nu_1) - (\lambda - 1)\theta_2(1 - \log\theta_2)$$

So if $\log\theta_2 > 1$ we can choose any $\lambda > 1$. In the other case we have to restrict our choice of $\lambda$ to get:

$$\lambda < 1 + \frac{\mathrm{KL}(\nu_2, \nu_1)}{\theta_2(1 - \log(\theta_2))}$$

So with an appropriate choice for $\lambda$ Poisson distributions are balanced with the same argument that makes Bernoulli distributions balanced.

### G.3 The Gaussian Distribution is Balanced

For the Gaussian distribution we leverage the fact that both the PDF and CDF of any Gaussian distribution can be expressed with the PDF and CDF of the standard normal distribution. With such decomposition, we can express $\alpha(M, j)$ as a function of these CDF/PDF and use some properties of the normal distribution.

We use the notations $f$ and $G$ for the PDF and CDF of the $\mathcal{N}(0,1)$ distribution, $\Delta$ for the gap between the two arms, and compute the expectation:

$$\alpha(M, j) = \int_{-\infty}^{+\infty} f_{1,j}(x) G_{2,j}(x)^M dx$$

$$\leq \int_{-\infty}^{z} f_{1,j}(x) G_{2,j}(x)^M dx + G_{2,j}(z)^M, \forall z \in \mathbb{R}$$

$$\leq \int_{-\infty}^{z} f\left(\frac{x - \mu_{1,j}}{\sqrt{j}}\right) G\left(\frac{x - \mu_{2,j}}{\sqrt{j}}\right)^M dx + G_{2,j}(z)^M$$

$$\leq \int_{-\infty}^{\frac{z - \mu_{2,j}}{\sqrt{j}}} f\left(y - \sqrt{j}\Delta\right) G(y)^M dy + G_{2,j}(z)^M$$

At this step we use two things: 1) the normal distribution satisfies $f(x - a) = e^{-a^2 + 2ax} f(x)$ for all $a, x$, and 2) $h : x \to (M+1) f(x) G(x)^M$ is a probability distribution of CDF $x \to 1 - G(x)^{M+1}$. We continue the computation with:

$$\alpha(M, j) \leq \frac{e^{-j\Delta^2}}{M+1} \int_{-\infty}^{\frac{z - \mu_2 j}{\sqrt{j}}} e^{\sqrt{j}\Delta y} h(y) dy + G_{2,j}(z)^M$$

$$\leq \frac{e^{-j\Delta^2}}{M+1} e^{\sqrt{j}\Delta \frac{z - \mu_2 j}{\sqrt{j}}} (1 - G\left(\frac{z - \mu_2 j}{\sqrt{j}}\right)^{M+1}) + G\left(\frac{z - \mu_2 j}{\sqrt{j}}\right)^M$$

$$\leq \frac{e^{-j\Delta^2}}{M+1} e^{\sqrt{j}\Delta \frac{z - \mu_2 j}{\sqrt{j}}} + G\left(\frac{z - \mu_2 j}{\sqrt{j}}\right)^M$$

As the inequality is true for all $z \in \mathbb{R}$, it holds that,

$$\forall y \in \mathbb{R}, \quad \alpha(M, j) \leq \frac{e^{-j\Delta^2}}{M+1} e^{\sqrt{j}\Delta y} + G(y)^M .$$

Now let $y_M$ be such as $G(y_M) = 1 - \frac{1}{\sqrt{M}}$. This value ensures that the second term satisfies $G(y_M)^M \leq e^{-\sqrt{M}} = o(M^{-2})$. Observe that $y_M = F^{-1}(\frac{1}{\sqrt{M+1}})$. Using the following equivalent of the quantile function of the normal distribution when the quantile is small (see for instance [27]):

$$F^{-1}(p) = -\sqrt{\log\frac{1}{p^2} - \log\log\frac{1}{p} + \log 2\pi} + o_{p\to0}(1) ,$$

there exists a constant $C \in \mathbb{R}$ such that $y_M \leq -C\sqrt{\log M - \log\log M + \log 4\pi}$. This yields

$$\alpha(M, j) \leq \frac{e^{-j\Delta^2}}{M+1} e^{-C\sqrt{j}\Delta\sqrt{\log M - \log\log M + \log 4\pi}} + e^{-\sqrt{M}}$$

Noting that for all $k \in \mathbb{N}^*$,

$$k \log\log M = o(C\sqrt{j}\Delta\sqrt{\log M - \log\log M + \log 4\pi})$$

we get that $\forall k \in \mathbb{N}^*$:

$$\alpha(M, j) = o\left(\frac{e^{-j\Delta^2}}{(M+1)(\log M)^k}\right)$$

This is sufficient to prove that the Gaussian distribution is balanced. Indeed, as for the Bernoulli distribution this term sums as a convergent geometric series in $j$, and with $M = O(t/(\log t)^2)$ we can make the sum in $t$ a convergent Bertrand Series.

### G.4 The Exponential Distribution is Not Balanced

For $j = 1$, a direct calculation yields

$$\alpha(M, 1) = \frac{1}{1 + \left(\frac{\mu_1}{\mu_2}\right)M}.$$

Using this, we now prove that the series in Assumption 2. of Theorem 3.1 is in $\Omega(\log(T))$, hence the balance condition is not satisfied. As all the $\alpha_k(M, j)$ are positive, one can write

$$
\begin{aligned}
\sum_{t=1}^{T}\sum_{j=1}^{\lfloor(\log t)^2\rfloor} \alpha_k(\lfloor \beta t/(\log t)^2 \rfloor, j) &\geq \sum_{t=1}^{T} \alpha_k(\lfloor \beta t/(\log t)^2 \rfloor, 1) \\
&= \sum_{t=2}^{T} \frac{1}{1 + \left(\frac{\mu_1}{\mu_k}\right)\lfloor \beta t/(\log t)^2\rfloor} \\
&\geq \sum_{t=2}^{T} \frac{1}{1 + \left(\frac{\mu_1}{\mu_k}\right)\beta t/(\log t)^2} \\
&\geq C\sum_{t=2}^{T} \frac{1}{t} = \mathcal{O}(\log(T)),
\end{aligned}
$$

where $C$ is some small enough constant that depend on $\mu_1, \mu_k$ and $\beta$.

## H Sketch of Proof with Forced Exploration

In this section, we explain how the proof of Theorem 3.1 is modified when we add forced exploration with $f_r = \sqrt{\log r}$, that is when in every round $r+1$ we add to $\mathcal{A}_{r+1}$ every arm $k$ such that $N_k(r) \leq f_r$. It is easy to verify that the proof of Lemma 4.1 remains unchanged, as it is inspired by the analysis of SSMC which also uses forced exploration. We now explain how forced exploration modifies the proof of Lemma 4.3 and how we upper bound the resulting new terms for any exponential family.

### H.1 Handling Forced Exploration in Lemma 4.3

The idea is to use the same proof sketch as without forced exploration. We note $f(r) = \sqrt{\log r}$ the forced exploration rate and $f^{-1}(r) = \exp(r^2)$ its inverse function.

Let us consider the round $a_r = f^{-1}(f(r) - 1)$. At this round, the value of exploration function is $f(r) - 1 = \sqrt{\log r} - 1$, which means that the number of pulls of arm 1 is at least $\lfloor\sqrt{\log r} - 1\rfloor \geq \sqrt{\log r} - 2$.

Now we look at the length of the interval $r - a_r$:

$$
\begin{aligned}
r - a_r &= r - f^{-1}(f(r) - 1) \\
&= r - \exp((\sqrt{\log r} - 1)^2) \\
&= r - \exp(\log r - 2\sqrt{\log r} + 1) \\
&= r(1 - \exp(-2\sqrt{\log r} + 1)) \\
&\sim r \text{ when } r \to +\infty
\end{aligned}
$$

As $r - a_r$ is equivalent to $r$ when $r$ is large, for any constant $\gamma > 0$ there exists a round $r_\gamma$ such that for $r > r_\gamma$: $r - a_r > \gamma r$. This means that after the round $a_r$ arm 1 faces a linear amount of duels, and has an history of at least $j = \lfloor \sqrt{\log r} - 1 \rfloor$ samples. Introducing $b_r$ the random variable giving the first time when $N_1(b_r) = \lfloor \sqrt{\log r} - 1 \rfloor$, we necessarily have $b_r \le a_r$. We now use that $N_1(r) \le (\log r)^2 \Rightarrow \sum_{j=1}^{\lfloor (\log r)^2 \rfloor - 1} \tau_j \le r$, which further implies

$$
b_r + \sum_{j=\sqrt{\log r}-1}^{\lfloor (\log r)^2 \rfloor - 1} \tau_j \le r \Rightarrow \sum_{j=\sqrt{\log r}-1}^{\lfloor (\log r)^2 \rfloor - 1} \tau_j \le r - b_r
$$

We can then use the same proof as in Appendix E:

$$
\begin{aligned}
\mathbb{P}\left(N_1(r) \le (\log r)^2\right) &\le \mathbb{P}\left(\exists j \in \{\lfloor \sqrt{\log r} - 1 \rfloor, ..., \lfloor (\log r)^2 \rfloor - 1\} : \tau_j \ge \frac{r - b_r}{(\log r)^2 - \lfloor \sqrt{\log r} - 1 \rfloor}\right) \\
&\le \mathbb{P}\left(\exists j \in \{\lfloor \sqrt{\log r} - 1 \rfloor, ..., \lfloor (\log r)^2 \rfloor - 1\} : \tau_j \ge \frac{r - a_r}{(\log r)^2 - \lfloor \sqrt{\log r} - 1 \rfloor}\right) \\
&\le \mathbb{P}\left(\exists j \in \{\lfloor \sqrt{\log r} - 1 \rfloor, ..., \lfloor (\log r)^2 \rfloor - 1\} : \tau_j \ge \frac{r - a_r}{(\log r)^2}\right) \\
&\le \mathbb{P}\left(\exists j \in \{\lfloor \sqrt{\log r} - 1 \rfloor, ..., \lfloor (\log r)^2 \rfloor - 1\} : \tau_j \ge \frac{\gamma r}{(\log r)^2}\right)
\end{aligned}
$$

The constant $\gamma$ does not change the sketch of proof, and we finally have:

$$
\sum_{r=1}^{T} \mathbb{P}(N_1(r)) \le (\log r)^2) \le r'_K + \sum_{r=r'_K}^{T} \sum_{j=\lfloor f_r \rfloor - 1}^{(\log r)^2} \left[ (K-1)\mathbb{P}(X_{c_r, c_r, j} < M_{r,j}) + \sum_{k=2}^{K} \alpha_k(M_{r,j}, j) \right] \quad (11)
$$

for any sequence $M_{r,j}$, and a new constant $r'_K$. Observe that the sum in $j$ does not start in 1 as it does in the statement of Lemma 4.3 in the absence of forced exploration. This justifies the introduction of the *generalized balance condition* in Appendix H.2

## H.2 Exponential Families Satisfy a Generalized Balanced Condition

To conclude the proof as in Theorem 3.1, as Random Block Sampling satisfies the Diversity Property, from (11) (with the choice $M_{r,j} = \lfloor \beta r/(\log r)^2 \rfloor$) it is sufficient to prove that one-dimensional exponential families satisfy the following generalized balance condition.

**Definition** (generalized balance condition). *If* SDA *is defined with a forced exploration rate $f_r$ then the generalized balance condition for the rate $f_r$ is:*

$$
\forall \beta \in (0, 1), \quad \sum_{r=1}^{T} \sum_{j=\mathbf{f_r}}^{\lfloor (\log r)^2 \rfloor} \alpha_k(\lfloor \beta r/(\log r)^2 \rfloor, j) = o(\log T) .
$$

The following lemma proves that this holds in particular for the choice $f_r = \sqrt{\log(r)}$, which permits to prove that RB-SDA with this forced exploration sequence is asymptotically optimal for distributions that belong to any one-dimensional exponential family.

**Lemma H.1** (Generalized balance condition on exponential families). *If the exploration rate $f_r$ satisfies $\frac{f_r}{\log \log r} \to +\infty$ then any exponential family of distributions with one parameter satisfies the generalized balance condition.*

*Proof.* A distribution that belong to a one-dimensional exponential family has a density $f_\theta(y) = f(x,0)e^{\eta(\theta)y - \psi(\theta)}$ for some natural parameter $\theta \in \mathbb{R}$.

We observe that for any $y_1, ..., y_j \in \mathbb{R}^j$, if $\sum_{i=1}^j y_i \leq \mu_k$:

$$\prod_{u=1}^j f_{\theta_1}(y_u) = \prod_{u=1}^j e^{(\eta(\theta_1) - \eta(\theta_k))y_u - (\psi(\theta_1) - \psi(\theta_k))} f_{\theta_k}(y_u) \leq e^{-jI_1(\mu_k)} \prod_{u=1}^j f_{\theta_k}(y_u)$$

This inequality ensures that for all $x, u \in \mathbb{R}$, if $F_{k,j}^{-1}(u) \leq \mu_k$:

$$F_{1,j}(x) \leq e^{-jI_1(\mu_k)} F_{k,j}(x) \Rightarrow F_{1,j}(F_{k,j}^{-1}(u)) \leq e^{-jI_1(\mu_k)} u$$

So for exponential families a strictly positive gap between two distributions leads to an exponential decrease of the ratio of the CDF of the sum. If we use the fact that for all $u \in \mathbb{R}$:

$$\alpha_k(M, j) = \int_{-\infty}^{+\infty} f_{1,j}(x) G_{2,j}(x)^M d\mathbb{P}(x)$$
$$\leq \int_{-\infty}^u f_{1,j}(x) G_{2,j}(x)^M + \int_u^{+\infty} f_{1,j}(x) G_{2,j}(x)^M d\mathbb{P}(x)$$
$$\leq F_{1,j}(u) + G_{2,j}(u)^M$$

Then $\forall \beta \in (0,1)$ and for all sequence $u_r$:

$$\sum_{r=1}^T \sum_{j=\mathbf{f_r}}^{\lfloor (\log r)^2 \rfloor} \alpha_k(\lfloor \beta r/(\log r)^2 \rfloor, j) \leq \sum_{r=1}^T \sum_{j=\mathbf{f_r}}^{\lfloor (\log r)^2 \rfloor} (1 - u_r)^{\lfloor \beta r/(\log r)^2 \rfloor} + e^{-jI_1(\mu_k)} u_r$$
$$\leq \sum_{r=1}^T \log(r)^2 (1 - u_r)^{\lfloor \beta r/(\log r)^2 \rfloor} + \sum_{r=1}^T \frac{e^{-f_r I_1(\mu_k)}}{1 - e^{-I_1(\mu_k)}} u_r$$

We now choose $u_r$ of the form $u_r = \frac{(\log r)^k}{r}$. Indeed, for the first term we get:

$$\log(r)^2 (1 - u_r)^{\lfloor \beta r/(\log r)^2 \rfloor} \leq \log(r)^2 \exp\left(-\lfloor \beta r/(\log r)^2 \rfloor \frac{(\log r)^k}{r}\right)$$
$$\leq \log(r)^2 \exp\left(-(\beta r/(\log r)^2 - 1)\frac{(\log r)^k}{r}\right)$$
$$\leq \zeta_k (\log r)^2 \exp\left(-\beta(\log r)^{k-2}\right)$$
$$= o(r^{-1}) \text{ for } k > 2$$

where $\zeta_k$ is an upper bound for $\exp(\frac{(\log r)^k}{r})$. From now on we work with $k = 3$. For the second term we have to study $u_r e^{-I_1(\mu_k) f_r}$:

$$u_r e^{-I_1(\mu_k) f_r} = \exp\left(\log u_r - I_1(\mu_k) f_r\right)$$
$$= \exp\left(3 \log \log r - \log r - I_1(\mu_k) f_r\right)$$

We see that $u_r e^{-I_1(\mu_k) f_r} = o(r^{-1})$ if $3 \log \log r - I_1(\mu_k) f_r \to 0$. This condition is achieved if $f_r / \log \log r \to +\infty$, hence an exploration rate satisfying this condition ensures the generalized balance condition for any exponential family of distributions with one parameter for this rate. $\quad\square$

We point out the fact that this forced exploration is not necessary in SDA, as we proved that some distributions (Bernoulli, Gaussian, Poisson) directly satisfy the balance condition defined in Assumption 2. of Theorem 3.1. We leave for future research an in-depth analysis of the properties of different families of distribution that could exhibit general conditions for the use of forced exploration (or not) in the SDA family.