[Reviews · NeurIPS 2020]

Review 1

Summary and Contributions: Several authors have tried to mimic the idea underlying Thompson sampling by replacing posterior sampling with bootstrap sampling. Unfortunately, when little data has been collected, subsampling does not adequately reflect the range of possible uncertainty. These algorithms require some extra randomization to succeed. This paper proposes a modification of this approach. They sample only from the observations of the 'leading' arm and compare the subsampled mean estimate to the empirical mean of challenger arms. All victorious arms are sampled and the algorithm proceeds to the next round. The paper shows that for several common distributions the algorithm attains the Lai Robbins lower bound on regret. Numerical experiments show very competitive performance.

Strengths: There have been so many papers about MABs with independent arms. It is hard to have really creative and elegant new ideas. In my opinion, the algorithms in this paper represent that. Moreover, the algorithms are simple enough that I could imagine them being used in practice. Beyond this, it seems generally to be well written, technically sound etc.

Weaknesses: For my taste, the numerical experiments in the body of the paper should be focused on problem settings that are more practical. (Not 2 arms with tens of thousands of observations). The regret analysis is already compelling in that regime. The numerical experiments in the appendix were more convincing to me.

Correctness: I did not find any mistakes, but I did not verify all of the proofs in the appendix.

Clarity: Yes. Thank you to the authors.

Relation to Prior Work: I am not an expert on the sampling based nonparametric bandit algorithms of the last 2 years, so the authors could have overlooked something I did not catch. But they certainly credit the influence of past work on this paper and clearly discuss differences.

Reproducibility: Yes

Additional Feedback: I got very stuck trying to understand what happens when some arms have been sampled zero times. What do we use for an empirical mean? I imagine many choices work for the theory, but please clarify what you use in the code. (Perhaps I missed this)


Review 2

Summary and Contributions: -The paper proposes a non-parametric multi-armed bandit algorithm based on 2 core ideas: sub-sampling and dueling. -The new method combines ideas from 2 previous works; it uses a sub-sampling scheme independent of the observations, and performs the duels in rounds between the leader arm and all the other arms. The independence of the sub-sampling scheme enables the use of concentration result for sub-samples, and the duels in rounds enable generalization to many arms. -The resulting algorithm is simple to implement, does not require distribution-specific tuning (as in parametric bandit algorithms), does not require wasteful forced exploration, and guarantees asymptotically optimal performance for a large class of exponential distributions.

Strengths: The proposed method is simple to implement, and can be applied to various reward distributions (including those with unbounded support) without the knowledge on the parametric form of the distribution. At the same time, it guarantees asymptotically optimal performance. As compared to parametric methods, the proposed method would be much more useful in practice as it is usually hard to verify the parametric assumptions and to tune hyperparameters in the online learning setup.

Weaknesses: The authors show that the proposed algorithm is asymptotically optimal when rewards follow a Gaussian distribution with "known" variance, and also present experiment results for this case. However, it is unclear how this "known" variance is incorporated in the algorithm. The presented Algorithm 1 does not require any input. The authors should clarify on this point.

Correctness: The claims and methods are stated in a clear manner.

Clarity: The paper is well-motivated, and the exposition is clear.

Relation to Prior Work: The authors did a thorough review of previous works, comparing them one by one with their method in terms of theoretical guarantee, empirical performance, assumptions, computational cost, robustness to distributions.

Reproducibility: Yes

Additional Feedback: ==== Comments after rebuttal ==== Thank you for the clarification.


Review 3

Summary and Contributions: The paper proposes the approach of sub-sampling duelling algorithm (SDA) to achieve efficient multi-armed bandit regret result that matches the asymptotic lower bound. The main feature of the SDA approach is that it is non-parametric, working for several families of distributions and it does not require to know the distribution families before hand. This approach seems to be different from the UCB approach and the Thompson sampling approach, the two dominant approaches in the MAB literature. It has the poential to provide alternative learning methods in practice.

Strengths: - The paper proposes the SDA approach, which is different from the dominant UCB and Thompson sampling approach, and has some potential to be come a new approach in practice. - The instantiation of the SDA approach, RB-SDA, provides the theoretical guarantee that matches the lower bound result. - Empirical evaluation in synthetic data sets show that the approach is promising.

Weaknesses: - One key problem that I did not find an answer in the paper is that whether the proposed algorithm still provides any regret guarantee if the unknown distribution does not follow the conditions given in Theorem 1. This is very important to understand the limitation of the approach. - The approach is not brand new, and it is built upon prior work [1] and [2]. In particular SSMC algorithm in [2] also uses the sub-sampling approach and also achieves the optimal result in a nonparametric way. The only difference seems to be that [2] requires a forced exploration but the current paper does not require that.

Correctness: The claims seem to be correct, though I did not check the complete proofs in the appendix.

Clarity: The paper is in general easy to follow, but some points need to be further clarified.

Relation to Prior Work: Yes.

Reproducibility: Yes

Additional Feedback: The authors propose the SDA approach, which has the potential to be an alternative to UCB and Thompson sampling for bandit learning. The main advantage of the proposed approach is that it is non-parametric, and for exponential families of distritions, it could achieve the optimal regret bound matching the lower bound, without knowing which distribution family the unknown distribution belongs to. However, I feel that achieve optimal regret guarantee is mostly theoretical interest. UCB may not be optimal in this sense but it could achieve consistent regret gurantee for all distributions. For the current approach, e.g. RB-SDA algorithm, it is unclear to me if its theoretical guarantee of O(log T) regret bound could be achieved for any distribution. Theorem 3.1 is only stated in the way that conditions 1 and 2 need to hold for the regret guarantee. What if 1 and 2 do not hold? Can we still achieve O(log T) regret (may not be optimal)? If SDA could achieve consistent O(log T) regret for all distributions, and optimal regret for the exponential family in a nonparameteric way, then I believe SDA would become an attractive alternative to UCB or Thompson sampling in practice. But if not, then I would only view SDA as of theoretical interest, because one cannot typically guarantee that the unknown distribution encountered in practice is of exponential family. I believe this is an important missing point in the paper, and a positive answer may significantly raise the contribution of the paper. The second issue of the paper is its novelty comparing to prior work, especially the SSMC algorithm from [2]. From the authors' description in the paper, it looks like SSMC in [2] already achieves nonparametric optimal regret bound for the exponential families of distributions. The main difference is that SSMC uses a deterministic sub-sampling that depends on the past rewards, while SDA framework proposed in the current paper uses random or deterministic sub-sampling that does not depends on the history. Also SSMC requires a forced exploration for arms that have very few samples, but SDA in this paper does not require forced exploration. But such difference looks to be minor. Thus the contribution of the paper seems to be minor. Other issues: - Line 30-31. It talks about optimal tuning, and in various places the papers talk about optimal regret. I think this needs to be more specific and needs more detailed discussion. USB already matches the asymptotic lower bound in the gap terms and the time horizon term T. So what is exactly the optimal regret? Meaning the matching of the constant term, or measuring the gap in terms of KL divergence? - Algorithm 1. The SP-SDA algorithm conducts a duel between the leader and every other arm. This makes it having a separate round concept. Is it necessary? Is it possible that the algorithm just chooses a random opponent in the arm set and play one duel between the leader and this opponent? Some discussion would be helpful. -line 139, "SSDA" appears here and in some other places, where in most places the authors use SDA. What is SSDA? Or is it just a typo? - line 161, "... that have non-overlapping support". The non-overlapping support is not very clear to me. Needs more elaboration. - Experiment section. UCB is not compared. I think it is important to compare against UCB, since UCB is the method used in many solutions. Moreover, different exploration parameter on the UCB confidence radius term should be used to see the effect comparing to the sub-sampling method. - Lines 256-257 The sentence explains the reason why the lower bound is above the upper bounds as the lower bound is asymptotic. But it still looks strange to me that the lower bound is significantly above all the upper bounds. Is it really just due to the asymptotic behavior or is there some other issue in the experiments? Perhaps the authors could run the experiments with a much higher time horizon to see if the lower bound will eventually go below the algorithm regrets. The figures 1 and 2 just look suspicious in this regard. --------- Post Response Comments ----------------- I am in general satisfied with the authors' response, and agree with the other reviewers that the paper is worth to be published at NeurIPS. The authors mention that some minor change would allow their algorithm to work with bounded distributions. I would very much like to see this research, and hope that the authors could add this part in their final version.


Review 4

Summary and Contributions: The authors propose a new algorithmic framework for multi-armed bandits which is based on resampling, and in particular sub-sampling and dueling (hence the SDA name). It is shown that this algorithm can be proven to match the Lai and Robbins under Gaussian, Bernoulli, and Poisson distributed arms with limited assumptions.

Strengths: The work appears quite strong theoretically, and provides an interesting framework for future work. It provides a new algorithmic setting with limited assumptions on the underlying distributions. Empirically the work is good and in that it backs up the theoretical work it seems quite strong and extensive. Note, though, that all algorithms compared against are quite close in comparison (as one might perhaps expect from their bounds). The authors also do a good job of relating to previous approaches, and the work is quite clear and the various Lemmas are well explained---although more on this below. The authors also plan to release code for this work and their experiments and have included it with the submission, which is great. Overall I found this to be a strong paper and worth accepting!

Weaknesses: What follows are a number of complaints/weaknesses of the work that center mainly around notational/descriptive clarity. But it is worth noting that I think these would make the paper better, but are otherwise somewhat minor complaints. The first is that while the work itself is quite clear, some of the notation is quite dense and terse. Some of this does obviously come from previous literature. For example, on line 95 we have Y_{k_i_r , N_{k_i_r}(r)} which although it does make sense can be difficult to unpack. I'm not sure how much this can be improved, but it would definitely be great to simplify in any way. Similarly, the paper could use some additional signposting of how results will be used later on in the work. For example, up to around line 111, it is unclear what form the sampling mechanism SP will take. This is given below, but having some brief description ahead of time would have made the algorithm easier to understand (with details obviously given later). Similarly, the more of a description and attempt to intuitively explain condition 2 of Theorem 3.1 would be quite useful to be given. These conditions come from the proof itself, but I would have liked to see more discussion of their implications. The authors do show how this is satisfied in the appendix for the distributions noted above, but I would have also liked to see slightly more discussion of this---albeit briefly as space is obviously a concern. Finally, I would have liked to see a greater discussion of the computational complexity would have been useful and/or additional results for longer horizons when this complexity can become more of a burden.

Correctness: As far as I can tell the work is correct (I have not fully read the proofs in the appendix however). The empirical results back this claim up and are relatively extensive.

Clarity: Yes, minus the (minor!) caveats given above.

Relation to Prior Work: As far as I can tell the authors situate this work quite well within previous work.

Reproducibility: Yes

Additional Feedback:

[Author Response · NeurIPS 2020]

We thank our four reviewers for their careful reading and their useful feedback. We provide answers below to the different questions raised by each reviewer. Numbered citations correspond to the bibliography of the paper.

**# Reviewer 1**  Regarding the initialization step, we will clarify that each arm is drawn once at the beginning of the algorithm in order to avoid empty history. Then, regarding the experiment section, we think that the frequentist experiments in the main text are interesting to check the theory and show that the logarithmic regret is achieved in a reasonable time. However, if the space limitation allows us to add the Bayesian experiments in the main text we would gladly do so in the camera-ready version of the paper (Table 7 and Table 8).

**# Reviewer 2**  Regarding the application of RB-SDA to Gaussian bandits with known variance, the variance parameter $\sigma$ is actually *not* needed as an input to the algorithm. Note however that our analysis only covers the case in which all arms belong to the same one-dimensional exponential family, i.e. in the Gaussian case it means that all arms should have the *same* variance. If the variances were to be different, a variant of the duels should be introduced, with some normalization by the variance for each arm, in spirit of what is proposed in the SSTC algorithm [2].

**# Reviewer 3**  First, we agree that providing a characterization of the distributions for which RB-SDA has logarithmic regret is an important future work. So far we have indeed only identified a sufficient condition in Theorem 1. We will emphasize this in our conclusion. Finding a bandit algorithm with logarithmic regret for as many distributions as possible is indeed a very interesting question. If by UCB you mean UCB1 of [5], this algorithm indeed provides regret guarantees for a large set of distributions: *sub-gaussian* distributions, and in particular *bounded* distributions. For the latter, we will add in the paper that RB-SDA can be turned to an algorithm with logarithmic regret for any bounded distribution by using the standard binarization trick proposed by [11] for Thompson Sampling. Moreover, in this work our focus is not only on logarithmic regret, or order-optimal regret (reasonable gap-dependent regret bound, as UCB1) but on asymptotically optimal regret (match the regret prescribed by the available lower bound). We will clarify this in the introduction, by providing the expression of the Lai and Robbins and Burnetas and Katehakis lower bounds.

Regarding the novelty with respect to the SSMC algorithm, we believe that deterministic history-dependent versus randomized history-independent sub-sampling is a significant difference. Indeed the philosophy of the algorithms are different: SSMC tries to disadvantage the leader by picking the smallest sub-sample (which requires a scan over all possible rolling means), while SDA uses only one sub-sample per duel, and the exploration is ensured by the diversity of the sub-samples encountered. We think that this idea may be more promising for extensions to problems in which 1) the leader may change a lot (e.g non stationary bandits), or 2) the duels are based on more costly estimators than the empirical mean (e.g. risk averse bandits), in which the computation of a "rolling" estimator will be computationally demanding in practice. One can also note that in our experiments randomized algorithms (RB-SDA, WR-SDA) tend to perform better than deterministic ones (SSMC and LB-SDA). Then, on the technical side, the novel elements of proof we developed for the analysis of randomized algorithms (such as Lemma 4.2) could be interesting for future work in this direction. Finally, we believe that understanding when we can get rid of forced exploration is conceptually important for multi-armed bandit algorithms. Indeed in more sophisticated settings such as structured bandits (see e.g. the OSSB algorithm of Combes et al. NeurIPS 2017), forced exploration can be quite harmful empirically.

The question of finding an alternative algorithm that is not round-based is very interesting. It is indeed a natural idea to choose a challenger at random instead of competing with all arms. However, we believe that this approach may not work when the leader is defined as the most sampled arm as in our setting. The intuition is the following: if the optimal arm is not competing in each round, then a sub-optimal leader with a sufficiently large lead could stay leader forever simply by beating other (possibly numerous) sub-optimal arms, even if it loses all its duels against the optimal arm. Still, we will check this intuition in practice as it may be interesting for future work.

Regarding the experimental section, we conducted all of our experiments with a larger set of algorithms including UCB1 and kl-UCB. However, we chose not to include these algorithms in the figures and tables as their regret was significantly worse than those of the algorithms we decided to represent (at least twice worse for UCB1 for instance). For this reason, we chose Thompson Sampling as the benchmark for "standard" bandit algorithms. Finally, the fact that the lower bound is much higher than the regret of good algorithms is always surprising, but it has already been observed frequently in the bandit literature, see e.g. [6,10].

**# Reviewer 4**  Thanks for your suggestions of improvements for the notation and presentation of the paper. Regarding the complexity of the algorithms, the question is a bit difficult because some algorithms (SSMC, LB-SDA) can be made efficient in rounds in which the leader doesn't change and no challenger arm has been pulled. Hence the problem is to estimate the computational cost of these algorithms in the other case, whose number of occurrences is problem-dependent. We will include a more extensive discussion on these issues in our revision.

[Meta-Review · NeurIPS 2020]

The panel of knowledgeable reviewers is clearly in favor of this paper and its results. The author response also helped significantly to address most concerns raised by the initial reviews. Hence, I am glad to recommend acceptance.